EMBO
reports

# Osteocalcin of maternal and embryonic origins synergize to establish homeostasis in offspring

Danilo Correa Pinto Junior[1,7], Isabella Canal Delgado[1,7], Haiyang Yang [2,7], Alisson Clemenceau[1,7], André Corvelo [3], Giuseppe Narzisi[3], Rajeeva Musunuri [3], Julian Meyer Berger[1], Lauren E Hendricks[1], Kazuya Tokumura[4], Na Luo[1], Hongchao Li[2], Franck Oury [5✉], Patricia Ducy [6✉], Vijay K Yadav [1✉], Xiang Li [2✉] & Gerard Karsenty [1✉]

## Abstract

**Many physiological osteocalcin-regulated functions are affected in adult offspring of mothers experiencing unhealthy pregnancy. Furthermore, osteocalcin signaling during gestation influences cognition and adrenal steroidogenesis in adult mice. Together these observations suggest that osteocalcin may broadly function during pregnancy to determine organismal homeostasis in adult mammals. To test this hypothesis, we analyzed in unchallenged wildtype and *Osteocalcin-deficient*, newborn and adult mice of various genotypes and origin maintained on different genetic backgrounds, the functions of osteocalcin in the pancreas, liver and testes and their molecular underpinnings. This analysis revealed that providing mothers are *Osteocalcin-deficient*, *Osteocalcin* haploinsufficiency in embryos hampers insulin secretion, liver gluconeogenesis, glucose homeostasis, testes steroidogenesis in adult offspring; inhibits cell proliferation in developing pancreatic islets and testes; and disrupts distinct programs of gene expression in these organs and in the brain. This study indicates that *osteocalcin* exerts dominant functions in most organs it influences. Furthermore, through their synergistic regulation of multiple physiological functions, osteocalcin of maternal and embryonic origins contributes to the establishment and maintenance of organismal homeostasis in newborn and adult offspring.**

**Keywords** Osteocalcin; Developmental Effect; Postnatal Physiology
**Subject Categories** Development; Metabolism

## Introduction

From energy metabolism to fertility and from musculoskeletal functions to behaviors, hormones regulate most physiological functions in adult mammals. As a result, they are essential contributors to the establishment and maintenance of organismal homeostasis (Drucker, 2018; Friedman, 2019; Kliewer and Mangelsdorf, 2019; Kreymann et al, 1987; Lo et al, 2014; Wu et al, 1995). The observation that children born from mothers who experienced an unhealthy pregnancy have a high propension to develop metabolic, endocrine, and neuropsychiatric diseases later in life (Burger et al, 1948; Hales and Barker, 2001) has long suggested that hormones made in the mother and crossing the placenta and/or made by the embryo might influence energy metabolism and other physiological processes in adult progenies (Chida et al, 2007; Choi et al, 2016; Karpac et al, 2005). Addressing this critical question of metabolism and endocrinology has been challenging because several hormones that could explain such an influence of gestation on postnatal energy metabolism and organismal homeostasis are either necessary for life (insulin), fertility (leptin), and/or are synthesized by the placenta (leptin, steroid hormones) (Accili et al, 1996; Ashworth et al, 2000; Hummel et al, 1966; Osinski, 1960; Oury et al, 2013b).

Osteocalcin is a peptide hormone that, following its binding to one of its three known receptors, regulates a broad array of physiological functions (Karsenty and Ferron, 2012; Karsenty and Olson, 2016; Pi et al, 2021; Pi et al, 2017). Those include glucose homeostasis, energy expenditure, exercise capacity, electrolyte homeostasis, blood pressure, male fertility, cognition, anxiety, and the acute stress response (Berger et al, 2019; De Toni et al, 2014; Glatigny et al, 2019; Gupte et al, 2014; Lee et al, 2007; Mao et al, 2021; Mera et al, 2016a; Oury et al, 2013b; Pi et al, 2016; Qian et al, 2021; Yadav et al, 2022). Remarkably, many physiological processes that are regulated by osteocalcin such as glucose homeostasis, cognition and even fertility are hampered in adult animals born from mothers experiencing an unhealthy pregnancy (Burger et al, 1948; Lee et al, 2007; Oury et al, 2013b; Oury et al, 2011). Furthermore, by signaling through the same

[1]Department of Genetics and Development, Vagelos College of Physicians and Surgeons, Columbia University, New York, NY, USA. [2]Guangdong Provincial Key Laboratory of Brain Connectome, Shenzhen-Hong Kong Institute of Brain Science-Shenzhen Fundamental Research Institutions, Shenzhen Institute of Advanced Technology, Chinese Academy of Sciences, 518055 Shenzhen, China. [3]New York Genome Center, New York, NY, USA. [4]Department of Bioactive Molecules, Pharmacology, Gifu Pharmaceutical University, Gifu, Japan. [5]INSERM U1151, Institut Necker Enfants-Malades (INEM), Université Paris Descartes-Sorbonne, Paris Cité, Paris, France. [6]Department of Pathology and Cell Biology, Vagelos College of Physicians and Surgeons, Columbia University, New York, NY, USA. [7]These authors contributed equally: Danilo Correa Pinto Junior, Isabella Canal Delgado, Haiyang Yang, Alisson Clemenceau. ✉E-mail: franck.oury@inserm.fr; pd2193@cumc.columbia.edu; vky2101@cumc.columbia.edu; xiang.li@siat.ac.cn; gk2172@cumc.columbia.edu

one of the three osteocalcin receptors, Gpr158, in the adrenal glands and developing brain, the embryonic and maternal (i.e., developmental) pools of osteocalcin regulate adrenal steroidogenesis and behaviors in adult offspring (Oury et al, 2013b; Yadav et al, 2022). When considered together, these observations raise the prospect that osteocalcin of maternal and/or embryonic origins may be a more significant and global regulator of physiological functions and organismal homeostasis in adult offspring than anticipated. Moreover, the facts that osteocalcin is not synthesized in the placenta but crosses it and that inactivation of *Osteocalcin* does not cause perinatal lethality or sterility (Oury et al, 2011) make this hormone a good candidate to assess, in a proof-of-principle study, to what extent the functions of hormones during gestation influence organismal homeostasis in adult animals. Moreover, the fact that osteocalcin of both maternal and embryonic origins contribute to post-natal physiology allows one to further refine the question, and through rigorous crosses to determine which functions of osteocalcin are of maternal origin and which ones are of embryonic origins (Oury et al, 2013b; Yadav et al, 2022).

We addressed this question through physiological, cellular, and molecular means in newborn and adult *Osteocalcin (Ocn)-deficient* mice of different genotypes or origins and maintained on diverse genetic backgrounds. This analysis revealed that if mothers are *Ocn* haplo-insufficient, haploinsufficiency for *Ocn* in embryos hampers cell proliferation in developing pancreas and testes, testes steroidogenesis, insulin secretion, liver gluconeogenesis and altogether glucose homeostasis in newborn and adult heterozygous offspring. Molecular analyses revealed that, providing that mothers are *Osteocalcin-deficient*, *Osteocalcin* haploinsufficiency in offspring affects in testes, liver and pancreas, where osteocalcin signals through the same receptor, Gprc6a, programs of gene expression that differ from those affected in the brain, an organ in which osteocalcin signals through different receptors. Furthermore, there is also an influence of embryonic osteocalcin on physiology postnatally. Hence, an interplay between *Osteocalcin* expression in mothers and embryos contributes to determine multiple physiological functions and homeostasis in offspring.

# Results

## Definition of the models and genotypes used in this study

To test whether maternal and/or embryonic osteocalcin exerts regulatory functions during pregnancy that affect physiology in adult offspring we analyzed *Osteocalcin (Ocn)*+/− mice born from either WT, *Ocn*+/− or *Ocn*−/− mothers using as negative controls WT mice born from WT mothers and as positive controls *Ocn*−/− mice born from *Ocn*+/− or *Ocn*−/− mothers (Fig. 1A). We also analyzed *Ocn*+/− mice born from WT mothers. Our reasoning in analyzing offspring of these various crosses was the following. First, if an osteocalcin-regulated function is affected in adult *Ocn*+/− offspring born from either *Ocn*+/− or *Ocn*−/− mothers, it shows that *Osteocalcin* haploinsufficiency suffices to hamper this function, in other words that this function is dominant. Second, if *Ocn*+/− mice born from *Ocn*+/− or *Ocn*−/− mothers display the phenotype and/or molecular perturbations of interests but *Ocn*+/− mice born from WT mothers do not, it indicates that

maternal but not embryonic osteocalcin determines in part the extent of this function in adult offspring. Third, if this osteocalcin-regulated function or molecular event is affected in *Ocn*+/− mice born from WT mothers, it reveals an influence of embryonic osteocalcin in setting up this function in adult mice. Fourth, if this osteocalcin-regulated function is equally affected in *Ocn*+/− mice whether they are born from *Ocn*+/− or *Ocn*−/− mothers, it indicates that a single allele of maternal *Ocn* is insufficient to rescue the deleterious consequences of *Ocn* haploinsufficiency in embryos on this function. The same is true if *Ocn*−/− mice born from *Ocn*+/− or *Ocn*−/− mothers have a phenotype of equal severity.

To reach conclusions of broad physiological significance, this analysis was performed in mice unchallenged by any diet manipulation and in up to three *Ocn-deficient* mouse strains generated through different means, maintained on distinct genetic backgrounds and studied at two different sites (Vagelos College of Physicians and Surgeons, New York City, USA; Chinese Academy of Science, Shenzen, China) (Fig. 1B). The first strain, *Ocn_GK*, was generated through homologous recombination in embryonic stem cells and has been maintained on a 129/SvEv genetic background since 1995 (Ducy et al, 1996). The genomic architecture of the *Osteocalcin* locus in this mouse strain has been previously reported (Ducy et al, 1996). The second strain termed *Ocn_BW* was generated by CRISPR-Cas9 genome editing on a mixed C57BL/6 J/C3H background, backcrossed twice on C57BL/6J background and then four times on a 129/SvEv background (Diegel et al, 2020). Hence, *Ocn_BW* mice are on a mixed 129/SvEv with components of C57BL/6J and C3H background. Analysis of energy metabolism in this strain was affected by the existence of an overt obesity restricted to WT mice (Figs. 1C and EV1A,B). A whole genome sequencing analysis initiated to identify molecular bases of this obesity revealed the presence of over 11,000 insertions or deletions that were over 50 base-pairs (bp) long, and of over a million smaller length variants in the genome of WT *Ocn_BW* mouse when compared to the one of a WT 129/SvEv mouse (Fig. 1D). Conceivably, any of these changes might contribute to the obesity of these mice. Hence, to insure that the results we would obtain when analyzing glucose homeostasis parameters in this strain will not be confounded by the obesity of WT mice, we used two types of control mice, WT *Ocn_BW* mice and WT129/SvEv mice. For all other physiological functions analyzed in this strain we used WT *Ocn_BW* mice as controls. The third strain, *Ocn_XL*, also generated by CRISPR-Cas9 genome editing, has been maintained on a C57BL/6J background for over 10 generations (Qian et al, 2021). The sequence of the *Osteocalcin* locus in this strain is available on the National Center for Biotechnology Information web site. As for the *Ocn_GK* strain, WT mice in this strain were lean and therefore used as controls (Qian et al, 2021). *Ocn_XL*−/− mice were analyzed in the Chinese Academy of Science for static metabolic tests. In each strain, the absence of circulating osteocalcin in *Ocn*−/− mice was verified (Figure EV1E–G).

Prior to initiate a broad and time-consuming phenotypic analysis we verified that regardless of their mode of generation, genetic background, or body weight, disruption of a function of osteocalcin could be detected in these three mutant mouse strains. Specifically, we found that as seen in *Ocn_GK* mice, spatial learning and memory were significantly decreased in adult *Ocn_BW*+/− (Fig. 1E) and *Ocn_XL*−/− (Fig. 1F) mice born from their respective *Ocn*−/− mothers when analyzed by a novel object recognition test (Oury et al, 2013b).

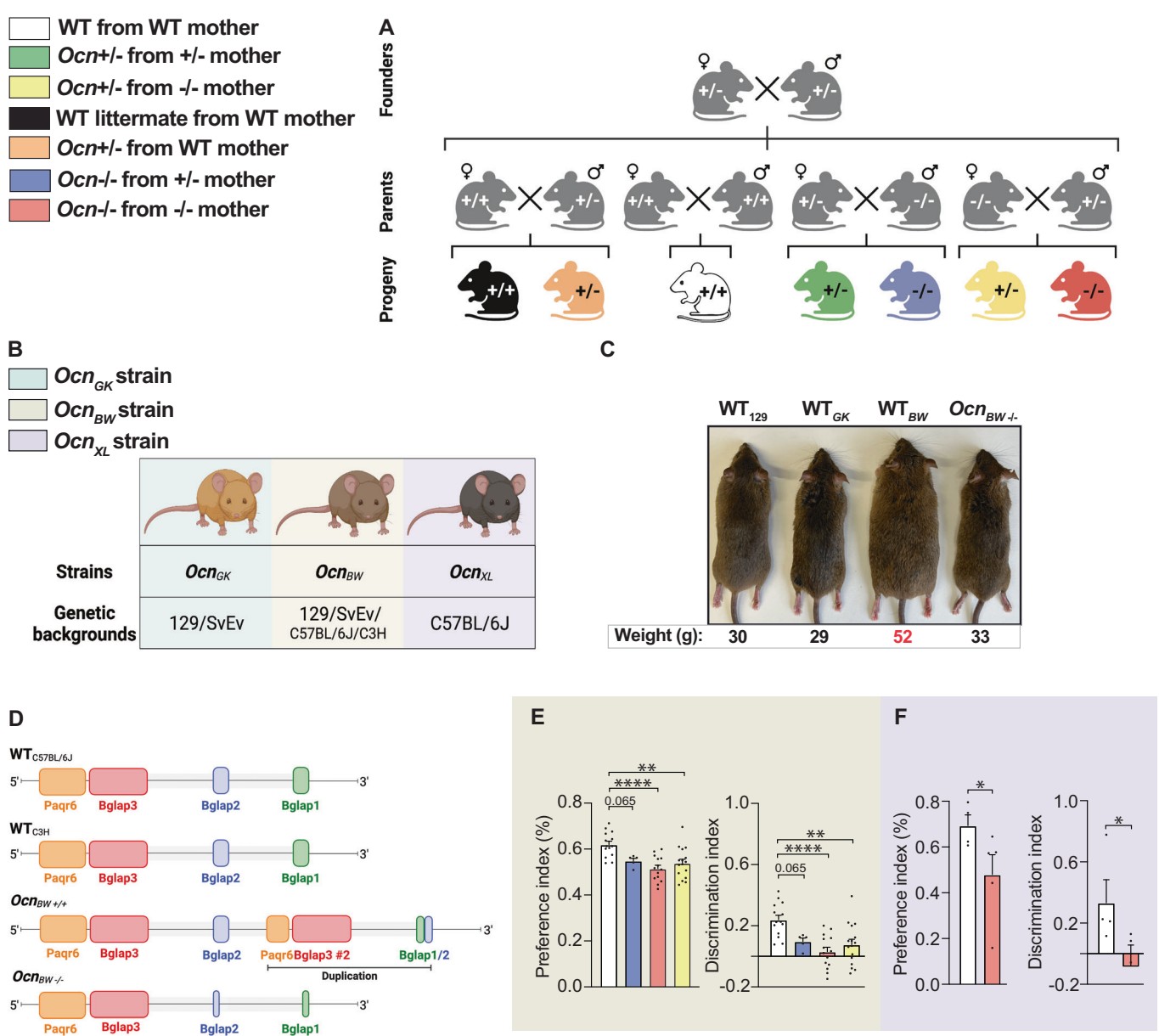

**Figure 1. Strategy used to determine the role of gestational osteocalcin in regulating adult physiological functions.**

(A) Schematic representation of the breeding strategy used for this study. $Ocn+/+$, $Ocn+/-$ or $Ocn-/-$ mothers were crossed with $Ocn+/+$, $Ocn+/-$, $Ocn-/-$ or $Ocn+/-$ fathers to generate progenies ($Ocn+/+$, $Ocn+/-$, or $Ocn-/-$) for analysis. (B) $Ocn$-deficient mouse strains ($Ocn_{GK}$, $Ocn_{BW}$ and $Ocn_{XL}$) and corresponding genetic backgrounds. (C) Photomicrographs of $Ocn$-deficient mouse strains and corresponding genetic backgrounds illustrating an obese phenotype in WT mice of $Ocn_{BW}$ strain. Body weight of corresponding genotypes is indicated below each photomicrograph. (D) Genome organization at the Ocn locus in WT mice of $Ocn_{BW}$ strain. (E) Preference index [left panel] and discrimination index in a novel object recognition test [right panel] in female mice of various genotypes of the $Ocn_{BW}$ strain. $n = 4$ or more mice per genotype analyzed. (F) Preference index [left panel] and discrimination index in a novel object recognition test [right panel] in female mice of various genotypes of the $Ocn_{XL}$ strain. $n = 3$ or more mice per genotype analyzed. Data information: In bar plots, each dot represents an individual mouse. All data are shown as mean ± SEM. Statistical significance was determined by one-way Kruskal–Wallis test followed by post hoc multiple comparisons test. $*p < 0.05$, $**p < 0.01$, $****p < 0.0001$. Source data are available online for this figure.

### The osteocalcin genotype of the mothers determines whether Osteocalcin haploinsufficiency in offspring hampers glucose homeostasis

The first physiological function we assayed to look for a possible influence of osteocalcin of maternal and/or embryonic origin on adult physiology was glucose homeostasis as assessed by the measurement of random fed and fasting blood glucose levels in adult (3-month-old) mice (Lee et al, 2007). We observed that random fed blood glucose levels were significantly higher in $Ocn_{GK}+/-$ mice born from $Ocn_{GK}+/-$ and $Ocn_{GK}-/-$ mothers than in control WT mice whereas $Ocn_{GK}+/-$ mice born from WT

mothers were normoglycemic (Fig. 2A,B). These data revealed that *Ocn* haploinsufficiency in offspring results in hyperglycemia only if their mothers also lack at least one allele of *Ocn*. When mothers were WT, the amount of maternal osteocalcin passing through the placenta sufficed to prevent any deleterious and long-lasting effect on glucose homeostasis post-natally that *Ocn* haploinsufficiency in offspring may have caused. In adult $Ocn_{GK}-/-$ mice, the hyperglycemia was also of similar severity whether their mothers were $Ocn_{GK}+/-$ or $Ocn_{GK}-/-$ (Fig. 2A,B), verifying that a single *Ocn* allele in mothers cannot overcome the deleterious consequences on glucose homeostasis in offspring resulting from the absence of *Ocn* expression in embryos.

This analysis was extended to other *Ocn-deficient* strains. As anticipated given their obesity, WT $Ocn_{BW}$ mice were hyperglycemic compared to WT 129 Sv/Ev mice and could not be used as controls (Figure EV1H,I). When WT129Sv/Ev mice bought from a vendor were instead used as controls, we observed that $Ocn_{BW}+/-$ mice born from $Ocn_{BW}+/-$ or $Ocn_{BW}-/-$ mothers were hyperglycemic whereas $Ocn_{BW}+/-$ mice born from WT mothers were not (Fig. 2C,D). Thus, in this mouse strain too, *Ocn* haploinsufficiency impairs glucose homeostasis in adult offspring only if their mothers are *Ocn-deficient*. As observed in the $Ocn_{GK}$ strain too, $Ocn_{BW}-/-$ mice showed equally high fasting and random fed blood glucose levels whether they were born from $Ocn_{BW}+/-$ or $Ocn_{BW}-/-$ mothers, illustrating that maternal osteocalcin cannot alleviate the deleterious consequences that the total absence of *Ocn* expression in embryos has on glucose homeostasis in adult offspring (Fig. 2C,D). In the third strain, the $Ocn_{XL}$-*deficient* mice (Qian et al, 2021), blood glucose levels were higher in $Ocn_{XL}-/-$ and $Ocn_{XL}+/-$ mice born from $Ocn_{XL}-/-$ mothers than in WT mice (Fig. 2E,F). These dominant abnormalities of glucose homeostasis were not observed when we studied fat mass. Indeed, $Ocn+/-$ mice of either strain tested did not harbor an increase in the weight of gonadal fat pads (Figure EV1C,D).

### The osteocalcin genotype of the mothers determines whether *Osteocalcin* haploinsufficiency in offspring hampers insulin secretion and β-cell proliferation

Osteocalcin regulates glucose homeostasis in part by signaling via its receptor Gprc6a in pancreatic β-cells to enhance insulin secretion and proliferation of these cells (Ferron et al, 2008; Lee et al, 2007; Pi et al, 2011; Sabek et al, 2015; Wei et al, 2014). Thus, we asked whether these two parameters are affected in *Ocn-deficient* offspring depending on the *Osteocalcin* genotype of the mother.

In both the $Ocn_{GK}$ and $Ocn_{BW}$ strains, fasting circulating insulin levels were significantly lower in adult $Ocn+/-$ mice whether they were born from $Ocn+/-$ or $Ocn-/-$ mothers compared to what it was in WT mice (Fig. 3A,B). In contrast and as observed for blood glucose levels, this decrease in circulating insulin levels was not observed if $Ocn+/-$ mice were born from WT mothers (Fig. 3A). These observations verified that *Ocn* haploinsufficiency in embryos affects circulating insulin levels in adult offspring only if their mothers are also *Ocn-deficient*. In both strains, circulating insulin levels were equally decreased in adult $Ocn-/-$ offspring whether they were born from $Ocn+/-$ or $Ocn-/-$ mothers (Fig. 3A,B). The fact that circulating insulin levels are decreased by half in

$Ocn_{BW}-/-$ mice provides a plausible element of explanation for why the obesity characterizing WT $Ocn_{BW}$ mice is not observed in mutant mice of these strain. In agreement with these results, glucose-stimulated insulin release was consistently hampered, albeit these differences did not reach statistical significance because of variability between samples, in $Ocn_{GK}+/-$ mice born from $Ocn_{GK}+/-$ or $Ocn_{GK}-/-$ mothers but not in those born from $Ocn_{GK}$ WT mice (Figs. 3A and EV2A). Adult $Ocn_{GK}-/-$ mice were equally poor glucose responder whether their mothers were $Ocn+/-$ or $Ocn-/-$ in both the $Ocn_{GK}$ and $Ocn_{BW}$ strains (Figs. 3C,D and EV2A,B).

To assess whether maternal osteocalcin influences the ability of adult *Ocn-deficient* offspring to secrete insulin because it affects β-cell endowment, we performed in vivo BrdU labeling in newborn mice (P1) of various genotypes and quantified β-cell proliferation by double insulin/BrdU immunohisto-fluorescence (IHF). We found that β-cell proliferation was significantly decreased in pancreata of $Ocn_{GK}+/-$ newborn mice compared to what was seen in pancreata of WT pups regardless of the genotype of the mother (Fig. 3E). β-cell proliferation was also significantly decreased in pancreata of $Ocn_{GK}-/-$ pups born from $Ocn_{GK}+/-$ or $Ocn_{GK}-/-$ mothers (Fig. 3E). Thus, as it is the case for blood glucose levels and insulin secretion, a single *Ocn* allele in the mother cannot rescue a defect in perinatal β-cell proliferation caused by *Ocn* haploinsufficiency (or complete absence) in the embryo.

### *Osteocalcin* genotype of the mother determines whether *osteocalcin* haploinsufficiency in offspring hampers liver gluconeogenesis

Osteocalcin also regulates glucose homeostasis through insulin-independent mechanisms. In particular, osteocalcin signals in hepatocytes through Gprc6a to promote liver gluconeogenesis, a process that is inhibited by insulin signaling (Pi et al, 2020). Therefore, we asked whether liver gluconeogenesis in *Ocn-deficient* offspring is affected by the genotype of the mother.

In both the $Ocn_{GK}$ and $Ocn_{BW}$ strains, adult $Ocn+/-$ born from WT mothers showed a similar increase of blood glucose levels as WT controls following a pyruvate challenge (Figs. 3F–I and EV2C,E). In contrast, when mothers were $Ocn+/-$ or $Ocn-/-$, the increase in blood glucose levels during a pyruvate challenge was lower in adult $Ocn+/-$ than in WT controls indicating that *Ocn* haploinsufficiency in the mother determines whether *Ocn* haploinsufficiency in the progeny will hamper liver gluconeogenesis (Fig. 3F–I). In contrast, as observed for blood glucose levels and insulin secretion, adult $Ocn-/-$ mice experienced a defect in pyruvate stimulated blood glucose levels of similar severity whether their mothers were $Ocn+/-$ or $Ocn-/-$ (Figs. 3F–I and EV2D,F–H).

To ascertain the existence of a developmental defect, we also assessed in newborn WT and *Ocn-deficient* mice born from mothers of different genotypes the expression in the liver of *Glycogen Phosphorylase* that encodes an enzyme that breaks down glycogen (*Pygl*) and of *Phosphoenolpyruvate carboxykinase* (*Pck1*) and *Glucose-6-phosphatase* (*G6pc*), two genes that are necessary for liver gluconeogenesis and whose expression is regulated by signaling through Gprc6a, the receptor for osteocalcin (Pi et al, 2011). Expression of these 3 genes was decreased in the liver of

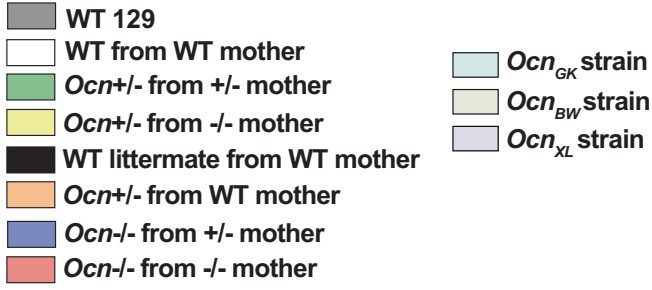

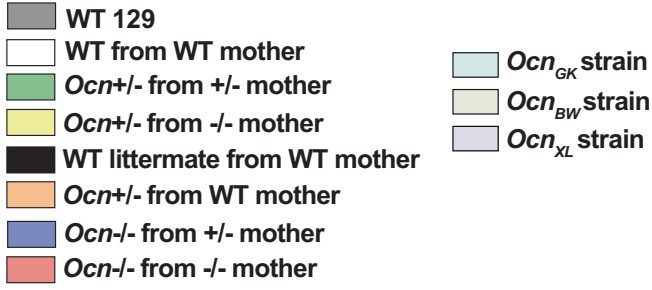

**Figure 2. Osteocalcin of developmental origin regulates blood glucose levels in adult mice.**

(A, C, E) fed and (B, D, F) fasting blood glucose levels in male and female mice from various genotypes of the $Ocn_{GK}$ (A, B). $n = 3$ or more mice per genotype analyzed. $Ocn_{BW}$ (C, D). $n = 3$ or more mice per genotype analyzed. $Ocn_{XL}$ strains (E, F). $n = 5$ or more mice per genotype analyzed. Data information: In bar plots, each dot represents an individual mouse. All data are shown as mean ± SEM. Statistical significance was determined by one-way Kruskal–Wallis test followed by post hoc multiple comparisons test. *$p < 0.05$, **$p < 0.01$, ***$p < 0.001$. Source data are available online for this figure.

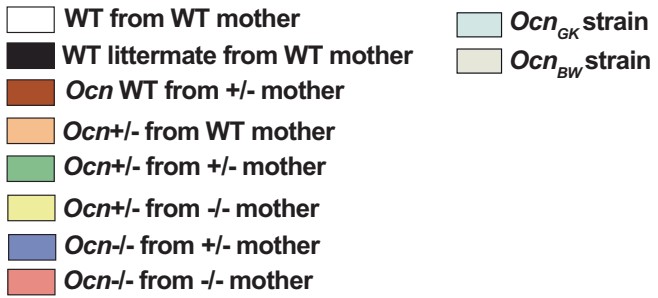

- □ **WT from WT mother**
- ■ **WT littermate from WT mother**
- ■ *Ocn* **WT from +/- mother**
- □ *Ocn*+/- **from WT mother**
- □ *Ocn*+/- **from +/- mother**
- □ *Ocn*+/- **from -/- mother**
- □ *Ocn*-/- **from +/- mother**
- □ *Ocn*-/- **from -/- mother**
- □ *Ocn*$_{GK}$ **strain**
- □ *Ocn*$_{BW}$ **strain**

**Figure 3.** Osteocalcin of developmental origin influences insulin secretion, b-cell proliferation and liver gluconeogenesis in newborn and adult mice.

(A, B) Fasting serum insulin levels in mice of different genotypes of the $Ocn_{GK}$ mouse strain (A). $n = 4$ or more mice per genotype analyzed. $Ocn_{BW}$ mouse strain (B). $n = 4$ or more mice per genotype analyzed. (C, D) Relative increases in serum insulin levels between baseline and 2 minutes after glucose administration in a glucose stimulated insulin secretion test in male mice of various genotypes in $Ocn_{GK}$ mouse strain (C). $n = 4$ or more mice per genotype analyzed. $Ocn_{BW}$ mouse strain (D). $n = 3$ or more mice per genotype analyzed. Values were normalized to WT values for comparison. (E) Relative β-cell proliferation in pancreas at postnatal day 1 (P1) pups of different genotypes in $Ocn_{GK}$ mouse measured after Insulin/BrdU immunostaining. Values were normalized to WT values for comparison. $n = 4$ or more mice per genotype analyzed. (F–I) Percent of initial blood glucose levels and corresponding area under the curve (AUC) in male (F, H) and female (G, I) mice of different genotypes in the $Ocn_{GK}$ mouse strain (F, G). $n = 3$ or more mice per genotype analyzed. $Ocn_{BW}$ (H, I) mouse strain following a Pyruvate tolerance test. $n = 4$ or more mice per genotype analyzed. (J) Relative Pck1, G6pc, and Pygl expression in the liver from P1 pups of different genotypes in the $Ocn_{GK}$ strain. $n = 5$ or more mice per genotype analyzed. (K) Pck1 and G6pc relative expression in the liver of P1 pups in the $Ocn_{BW}$ strain. $n = 4$ or more mice per genotype analyzed. Data information: In bar plots, each dot represents an individual mouse. All data are shown as mean ± SEM. Statistical significance was determined by one-way Kruskal–Wallis test followed by post hoc multiple comparisons test. *$p < 0.05$; **$p < 0.01$; ***$p < 0.001$; ****$p < 0.0001$, ns: not significant. Source data are available online for this figure.

$Ocn+/-$ and $Ocn-/-$ pups compared to WT only if mutant pups were born from $Ocn$-deficient mothers (Fig. 3J,K). These results confirmed that $Ocn$ haploinsufficiency in embryos hampers liver gluconeogenesis and glucose availability intracellularly in adult offspring only if their mothers lack at least one allele of $Ocn$. This decrease in gene expression was more severe in $Ocn+/-$ and $Ocn-/-$ pups when mothers were $Ocn-/-$ than when they were $Ocn+/-$ (Fig. 3J,K). We also observed that $G6pc$ expression and to a lesser extent $Pepck$ expression were decreased in postnatal day 1 (P1) $Ocn+/-$ pups born from WT mothers, an observation indicating that embryonic osteocalcin contributes to setting up the expression of these genes in newborn mice (Fig. 3K). The decrease in expression of these three genes was as or more severe in $Ocn+/-$ P1 pups born from $Ocn-/-$ mothers than in those born from $Ocn+/-$ mothers, but was normal in adult $Ocn+/-$ mice regardless of the genotype of their mothers (Compare Figs. 3J,K and EV2I). These results suggest that the abnormal gluconeogenesis observed in adult $Ocn+/-$ mice is due to a deficit in osteocalcin signaling during pregnancy.

When taken together, the analysis of these various aspects of glucose homeostasis in $Ocn$-deficient mice of various genotypes, origins, and genetic backgrounds provide convincing evidence that maternal osteocalcin is a determinant of pancreatic β-cell proliferation, insulin secretion, liver gluconeogenesis and overall, glucose homeostasis in adult mice provided it is delivered to embryos at WT levels.

### The osteocalcin genotype of the mother determines whether osteocalcin haploinsufficiency in offspring hampers testes steroidogenesis

Osteocalcin signaling through Gprc6a in Leydig cells of the testes promotes testosterone biosynthesis in cell culture and maintains normal circulating testosterone levels in adult male mice and humans (Oury et al, 2013a; Oury et al, 2011). At the time these data were collected, it was not known that $Osteocalcin$ haploinsufficiency in the mother may determine the consequence of $Osteocalcin$ haplo-insufficiency on physiology in the offspring (Oury et al, 2013a; Oury et al, 2011). In view of the results observed when studying various aspects of the regulation of glucose homeostasis by osteocalcin, this became an important question to address.

In both the $Ocn_{GK}$ and $Ocn_{BW}$ strains, circulating testosterone levels were significantly decreased in male adult $Ocn+/-$ mice born from $Ocn-/-$ mothers and to a lower extent in those born from $Ocn+/-$ mothers thus verifying that $Osteocalcin$ deficiency in mothers determines whether $Ocn$ haploinsufficiency in the adult

offspring hampers testes endocrine functions (Fig. 4A,B). The same was true for adult $Ocn-/-$ mice whether they were born from $Ocn+/-$ or $Ocn-/-$ mothers (Fig. 4A,B). To assess whether this effect of $Ocn$ haploinsufficiency in mothers and embryos on testes steroidogenesis originates in part from a defect of Leydig cell proliferation, we performed in vivo BrdU labeling in newborn mice and quantified Leydig cell proliferation on testes sections using analysis of $Steroidogenic Factor 1$ ($Sf1$) expression as a molecular marker of these cells. This analysis revealed that Leydig cell proliferation was markedly decreased in testes of $Ocn_{GK}+/-$ or $Ocn_{GK}-/-$ newborn mice regardless of the genotype of the mother (Fig. 4C). These data indicate that $Ocn$ haploinsufficiency in embryos alone hampers proliferation of Leydig cells.

Thus, to determine whether maternal osteocalcin also influences testes development in embryos, we measured expression of $Steroidogenic acute regulatory protein$ ($Star$) and $Cytochrome P450 family 11 subfamily A member 1$ ($Cyp11a1$), two genes that are necessary for testosterone biosynthesis and whose expression is regulated by $Osteocalcin$ (Oury et al, 2011), and of $Luteinizing hormone/choriogonadotropin receptor$ ($Lhcgr$), which encodes the receptor of the pituitary hormone luteinizing hormone (LH) that is the canonical regulator of testosterone biosynthesis in Leydig cells (Zirkin and Papadopoulos, 2018). We observed that the expression of $Star$ and $Cyp11a1$ were decreased in testes of newborn male $Ocn+/-$ pups born from $Ocn+/-$ or $Ocn-/-$ mothers but not in those born from WT mothers (Fig. 4D,E). These results support the notion that $Ocn$ expression in mothers and embryos are both necessary and synergize to regulate testes development and thereby of testosterone biosynthesis in offspring.

### Genetic pathways regulated by developmental osteocalcin in testes

The observation that osteocalcin of embryonic and maternal origins cooperates to determine the quality of physiological processes taking place in the pancreas, liver, and testes of adult offspring, raises the question of the molecular modes of action of osteocalcin during gestation. To begin addressing this question, we performed an RNA sequencing (RNA-seq) analysis in testes of newborn WT mice born from WT mothers and in those of newborn $Ocn+/-$ and $Ocn-/-$ mice born from $Ocn-/-$ mothers. Given that circulating testosterone levels were similarly decreased in $Ocn+/-$ mice in both the $Ocn_{GK-}$ and $Ocn_{BW}$-deficient mouse strains, this analysis was performed only in the former strain.

Consistent with results obtained when analyzing testes functions in $Ocn+/-$ and $Ocn-/-$ adult mice, a clustering of the most

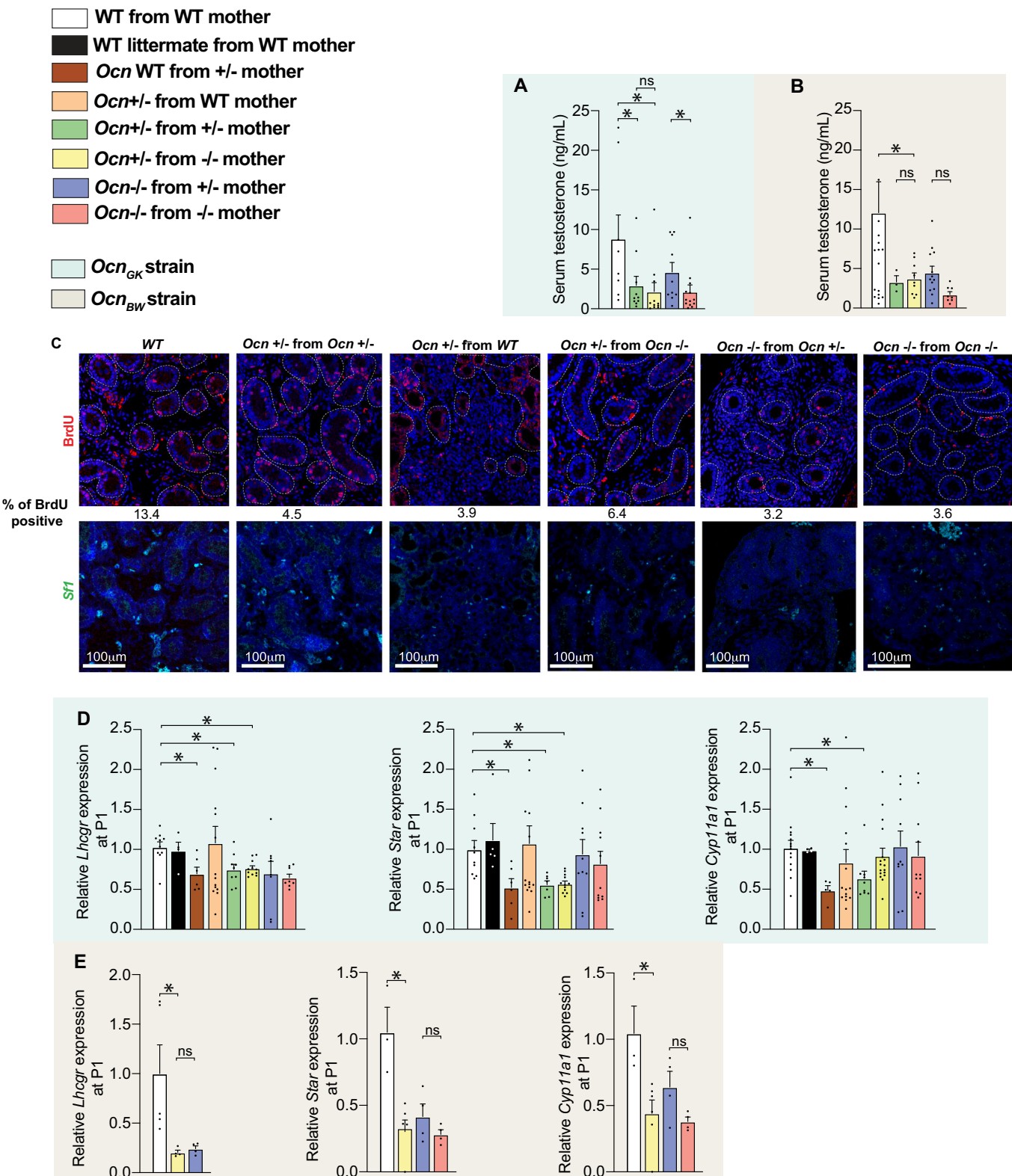

dysregulated genes in testes of Ocn+/− and Ocn−/− newborn mice compared to those of WT ones, failed to distinguish heterozygotes from homozygotes samples (Fig. 5A). Among the most down-regulated genes in testes of both Ocn+/− and Ocn−/− newborn

mice we found, as expected, Star and Lhcgr, two genes involved in testes steroidogenesis (Fig. 5B). Moreover, a gene ontology (GO) term analysis of the genes most significantly downregulated (FC < 0.5) identified a decrease in the expression of markers of

**Figure 4.  Osteocalcin of developmental origin influences testicular steroidogenesis in newborn and adult mice.**

(A, B) Relative testosterone levels in adult mice of different genotypes in $Ocn_{GK}$ mouse strain (A). $n = 8$ or more mice per genotype analyzed. $Ocn_{BW}$ (B) mouse strain. $n = 3$ or more mice per genotype analyzed. Results were normalized to WT levels for comparison. (C) BrdU immunostaining and quantification of % BrdU positive cells in testes at P1 in pups of different genotypes in $Ocn_{GK}$ [top panel] mouse strain. In situ hybridization analysis of $Sf1$ in testes at P1 pups of different genotypes in $Ocn_{GK}$ [bottom panel] mouse strain. (D, E) Relative $Lhcgr$, $Star$ and $Cyp11a1$ expression in testes at P1 of different genotypes in $Ocn_{GK}$ (D) mouse strain. $n = 4$ or more mice per genotype analyzed. $Ocn_{BW}$ (E) mouse strain. $n = 4$ or more mice per genotype analyzed. Data information: In bar plots, each dot represents an individual mouse. All data are shown as mean ± SEM. Statistical significance was determined by one-way Kruskal–Wallis test followed by post hoc multiple comparisons test. *$p < 0.05$; ns: not significant. Source data are available online for this figure.

hormonal activity in testes of both $Ocn+/-$ and $Ocn-/-$ newborn mice (GO:0005179) (Fig. 5C–E). Those include $Adrenomedullin$ ($Adm$) and $Natriuretic\ peptide\ C$ ($NppC$) that have been implicated in Leydig cell functions and testes development, respectively (Chan et al, 2008; Xia et al, 2007; Fig. 5C–E). We also observed that expression of $Period\ circadian\ protein\ homolog\ 1$ ($Per1$), a core gene involved in circadian rhythm, and of $Pck1$, a main control point for gluconeogenesis, was decreased in testes of $Ocn+/-$ and $Ocn-/-$ newborn mice (Figs. 5A,B and EV3A). In agreement with this latter observation, the GO term analyses based on the genes significantly downregulated (FC < 0.5) highlighted numerous pathways involved in glucose homeostasis such as glucose (GO:0006006), hexose (GO:0019318) and pyruvate (GO:0006090) metabolism (Figs. 5D and EV3B). Furthermore, a Kyoto Encyclopedia of Genes and Genomes (KEGG) analysis identified gluconeogenesis (mmu00010) as the most downregulated pathway in testes of both $Ocn+/-$ (stat.mean = −3.6; $p$ value = 3.9e-04, $q$-value = 3.4e-02) and $Ocn-/-$ newborn mice (stat.mean = −3.6; $p$ value = 2.2e-04, $q$-value = 5.0e-02) (Fig. 5F). Taken together these results indicate that, when mothers are $Ocn$-deficient, $Ocn$ haploinsufficiency in embryos disrupts testes steroidogenesis, gluconeogenesis, and possibly, the molecular clock in newborn mice.

## Developmental osteocalcin regulates similar genetic pathways in the liver, pancreas, and testes

The results of the transcriptomic analysis performed in testes provided us with the opportunity to ask whether the same genes are regulated by the interplay between maternal and embryonic osteocalcin in the pancreas and the liver, two organs where osteocalcin also signals through Gprc6a. Prior to performing this analysis, we verified by qPCR that the expression of a few genes identified as dysregulated by the transcriptomic analysis were indeed affected in testes of $Ocn$-deficient newborn mice. This survey showed that expression of $Per1$ and $Pck1$ was significantly decreased in testes of $Ocn-/-$ and $Ocn+/-$ pups whether they were born from $Ocn+/-$ or $Ocn-/-$ mothers (Fig. 6A,B). In contrast, the expression of these two genes was not decreased in testes of $Ocn+/-$ pups born from WT mothers (Fig. 6A,B). These results further establish that $Ocn$ haploinsufficiency in embryo affects programs of gene expression in testes only if mothers are $Ocn$-deficient too.

Having verified the validity of the RNA-Seq study in testes, we turned our attention to the pancreas and liver. We first asked through immunofluorescence whether the accumulation of components of the molecular clock or implicated in gluconeogenesis in pancreatic islets of newborn mutant mice was modified (Fig. 6C–E). The absence of an anti-Per1 antibody that could be used on paraffin sections prevented the analysis of its expression, but the levels of

Bmal1 and Clock, two other key components of the molecular clock, were lower in pancreatic islets of $Ocn+/-$ pups born from $Ocn+/-$ or $Ocn-/-$ mothers than in those of WT pups (Fig. 6C,D). For Bmal1, the decrease in its accumulation was not seen in $Ocn+/-$ pups born from WT mothers (Fig. 6C). The accumulation of Pck1 was also decreased in islets of $Ocn+/-$ newborn mice regardless of the genotype of their mothers (Fig. 6E). These results are consistent with the notion that embryonic and maternal osteocalcin synergize to influence the expression of key components of the molecular clock and of the gluconeogenic pathway in pancreatic islets in newborn mice.

In the liver, we relied on gene expression analysis and observed that the expression of $Per1$, $Bmal1$ and $Clock$, was significantly decreased in newborn $Ocn-/-$ and $Ocn+/-$ mice born from $Ocn-/-$ mothers (Fig. 6F). In contrast, only $Per1$ expression was significantly decreased in $Ocn+/-$ mice born from $Ocn-/-$ mothers suggesting a greater sensitivity of this gene's regulation to osteocalcin levels (Fig. 6F). Expression of these three genes was normal in the liver of $Ocn+/-$ newborn pups born from WT mothers, verifying that $Ocn$ haploinsufficiency in embryos affects liver gene expression in offspring only if their mothers were $Ocn$-deficient too (Fig. 6F). We also observed that the rhythmic expression of $Per1$, $Bmal1$, and $Clock$ was perturbed in the liver of adult $Ocn+/-$ mice born from $Ocn-/-$ mothers (Fig. 6G). These results are consistent with the notion that osteocalcin of maternal and embryonic origins synergizes to determine expression of the same key components of the molecular clock and gluconeogenic pathways in the liver, pancreas, and testes of newborn and adult offspring.

## Developmental osteocalcin regulates distinct genetic pathways in the brain

The results presented above raised the question as to whether osteocalcin regulates the same genes and genetic pathways in all organs independently of the receptor through which it signals. To address this question, we performed an RNA-seq analysis in the forebrain (including the hippocampus) where osteocalcin signals through Gpr158 and Gpr37 (Khrimian et al, 2017; Qian et al, 2021). Here again, this analysis was performed in newborn WT, $Ocn+/-$ and $Ocn-/-$ mice born from $Ocn-/-$ mothers.

As it is the case in the testes, a clustering analysis based on the identification of the most dysregulated genes, failed to distinguish any differences between forebrains of $Ocn+/-$ and $Ocn-/-$ pups (Fig. 7A,B). The expression of $Pck1$ that was downregulated in testes, pancreas and liver of newborn $Ocn$-deficient pups was normal in their forebrain (Figure EV4A). Expression of key components of the molecular clock was also similar in the forebrain and adult midbrain of $Ocn$-deficient and WT newborn

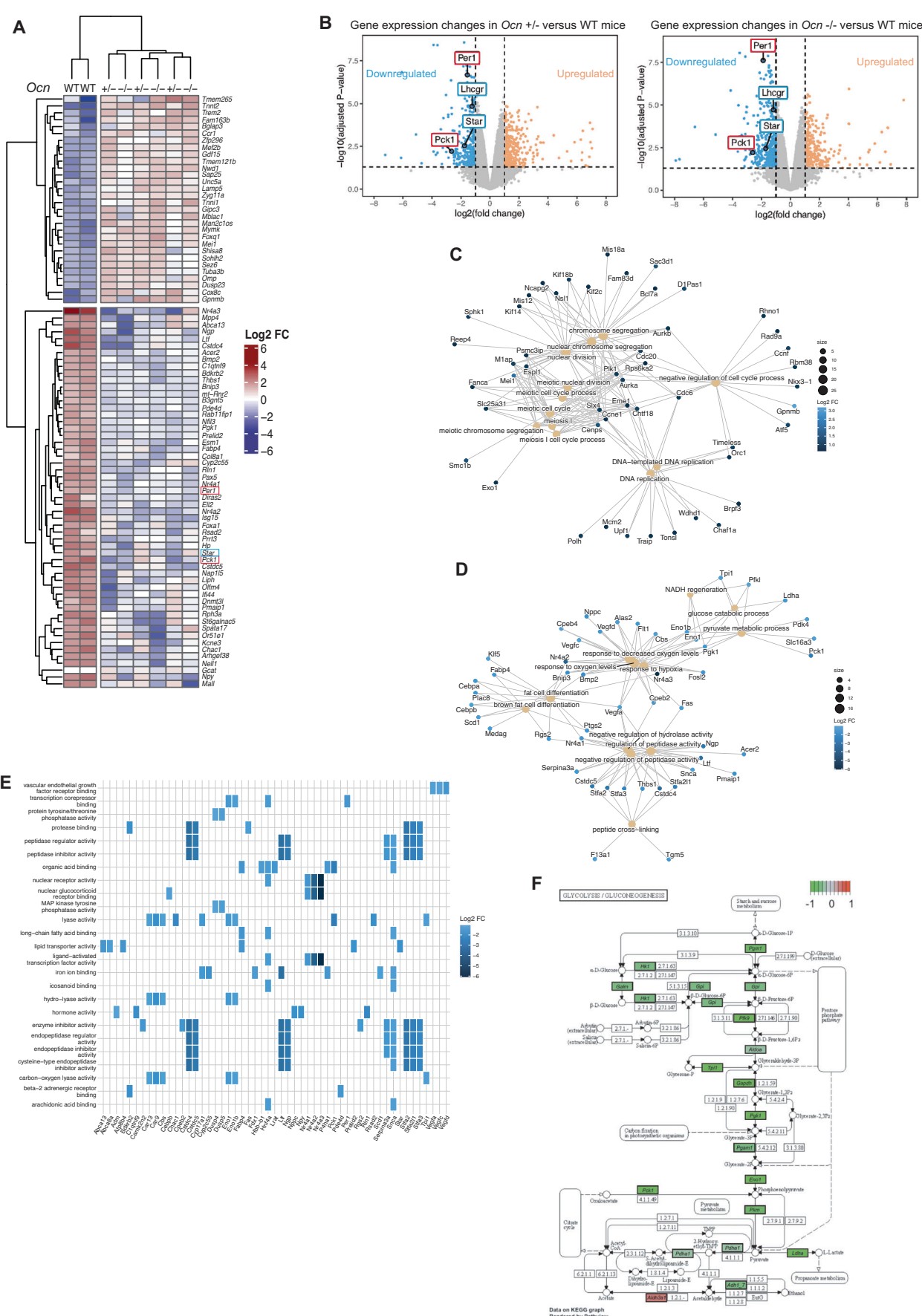

◄ **Figure 5. Transcriptomic analysis of osteocalcin signaling in testes of newborn mice of different genotypes.**

(A) Heatmap illustrating results of unsupervised hierarchical clustering of testes obtained from WT ($n = 2$), $Ocn+/−$ ($n = 3$) and $Ocn−/−$ ($n = 3$) mice. (B) Volcano plots of the genes differentially expressed in testes from $Ocn+/−$ [left panel] and $Ocn−/−$ [right panel] mutant compared to WT mice. Each dot represents one gene or pseudogene ($n = 3$). (C, D) Category netplots depicting the linkages between the top 12 biological processes (GO term) and genes that are significantly (C) upregulated or (D) downregulated in the testes of both $Ocn+/−$ and $Ocn−/−$ compared to WT mice. (E) Heatplot of the top 25 GO molecular functions (GO term) that are significantly downregulated in both $Ocn+/−$ and $Ocn−/−$ compared to WT mice. (F) KEGG pathview graph of glycolysis/gluconeogenesis pathways comparing the transcriptomic data obtained from $Ocn+/−$ mice to those from WT mice. Data information: Values close to −1 (green) indicate a downregulation of the gene in $Ocn+/−$ compared to WT, values close to 0 (gray) indicate no difference and values close to 1 (red) indicate an upregulation in $Ocn+/−$ compared to WT.

mice (Figure EV4A,B). These results suggested that osteocalcin regulates different genes and genetic pathways depending on the organ where it signals.

Consistent with this notion, we noticed that the most upregulated GO pathways identified through this analysis included those involved in neuronal death, and in inflammation such as the production of interleukin-1 and of the tumor necrosis factor superfamily (Fig. 7C,D). The same GO analysis also identified several genetic pathways involved in intracellular energy metabolism such as pyruvate metabolic process (GO:0006090), and most notably, glucose catabolism (GO:0006007) and canonical glycolysis (GO:0061621, all adjusted $p$ values <0.006, Fig. 7C). Importantly, virtually all genes downregulated in these specific pathways in the forebrain (35 out of 41 genes) were distinct from those involved in glucose homeostasis and downregulated in the testes (compare Figs. 5E and 7C). Taken together these results suggest that osteocalcin regulates different genes and genetic pathways in organs where it signals through Gprc6a and in those like the brain where it signals through Gpr158 or Gpr37.

## Discussion

The goals of this study were several. One of them was to determine whether functions of osteocalcin are dominant or recessive. A second one was to determine the respective importance of embryonic and maternal osteocalcin in defining homeostasis postnatally. To address these goals, we analyzed its functions and their molecular underpinnings in the liver, pancreas, testes and for the transcriptomic analysis, the brain of WT and *Ocn-deficient* unchallenged mice of various genotypes and origins. We made two observations. The first is that osteocalcin exerts dominant functions in the pancreas, liver, and testes as it does in the brain (Oury et al, 2013a) and that maternal and embryonic osteocalcin synergize to regulate multiple physiological processes and thereby contribute to establishing organismal homeostasis in adult offspring. This was observed regardless of the genetic background of the animals analyzed. A third observation obtained through a molecular analysis conducted in peripheral organs and in the brain is that osteocalcin affects distinct genetic pathways in peripheral organs where it signals through Gprc6a and in the brain where it signals through Grp158 and Gpr37.

Our analysis of the $Ocn_{BW}$ mouse strain generated results that differ from those reported or mentioned by Diegel et al (2020). Because of its exhaustive genomic, phenotypic, and physiological analyses, the present study, performed on a large number of mice at birth and in adulthood, fasting or fed, provides explanations for the apparent discrepancies between results published by Diegel et al (2020) and those published by others (Lee et al, 2007; Oury et al,

2013a; Pi et al, 2011; Pi et al, 2020). One explanation certainly resides in the small number of mice analyzed by Diegel et al (2020) and the presence of an overt hyperglycemia at baseline in their WT controls that had been overlooked. In addition to technical considerations, the massive genomic rearrangements observed in what should have been WT controls weakens the credibility of the original analysis of this mouse strain (Diegel et al, 2020).

Analyses of newborn and adult $Ocn+/−$ mice of various maternal origins and distinct genetic backgrounds show that developmental osteocalcin, i.e., maternal and embryonic, contributes to the regulation of pancreatic β-cell proliferation, insulin secretion, liver gluconeogenesis and altogether, glucose homeostasis in newborn or adult mice. Developmental osteocalcin also promotes testicular cell proliferation during development and testosterone biosynthesis in adult mice. That the regulation of several physiological processes in adult offspring significantly relies on osteocalcin signaling during gestation does not exclude that osteocalcin of postnatal origin contributes to the regulation of the same physiological functions as well as of others that do not seem to be affected by developmental osteocalcin. Indeed, we have previously shown that it is osteocalcin made postnatally that regulates muscle function during exercise (Berger et al, 2019; Mera et al, 2016a). Moreover, we verified here that the postnatal pool of osteocalcin regulates glucose homeostasis and testes steroidogenesis too. Thus, this work identifies two groups of osteocalcin functions, a first one is that most osteocalcin regulatory functions are dominant and fulfilled in part by osteocalcin signaling during gestation. A smaller group of physiological functions such as muscle function during exercise and the acute stress response that are regulated by osteocalcin of postnatal origin. In view of the observation that energy metabolism and organismal homeostasis in adult animals are regulated in part by osteocalcin acting during gestation, we believe it is likely that additional hormones signaling during pregnancy may contribute to the establishment of homeostasis in adult offspring.

Our results show that maternal and embryonic osteocalcin signaling during gestation synergize to influence physiological functions analyzed postnatally. Indeed, *Ocn* haploinsufficiency in the embryos affects physiology in peripheral organs and in the brain only if the mother is also *Ocn* haplo-insufficient or null (Oury et al, 2013b). These observations may have direct relevance to our understanding of the pathogenesis of some human pathological situations. Specifically, the deleterious influence that an unhealthy pregnancy has on energy metabolism, cognitive functions and other physiologies in adult offspring is reminiscent of phenotypes observed in animals lacking *Ocn* or its receptors. This suggests that a dysregulation in *Ocn* expression or circulating levels during a difficult or stressful pregnancy may affect physiology postnatally (Burger et al, 1948; Hales and Barker, 2001; Karsenty and Ferron,

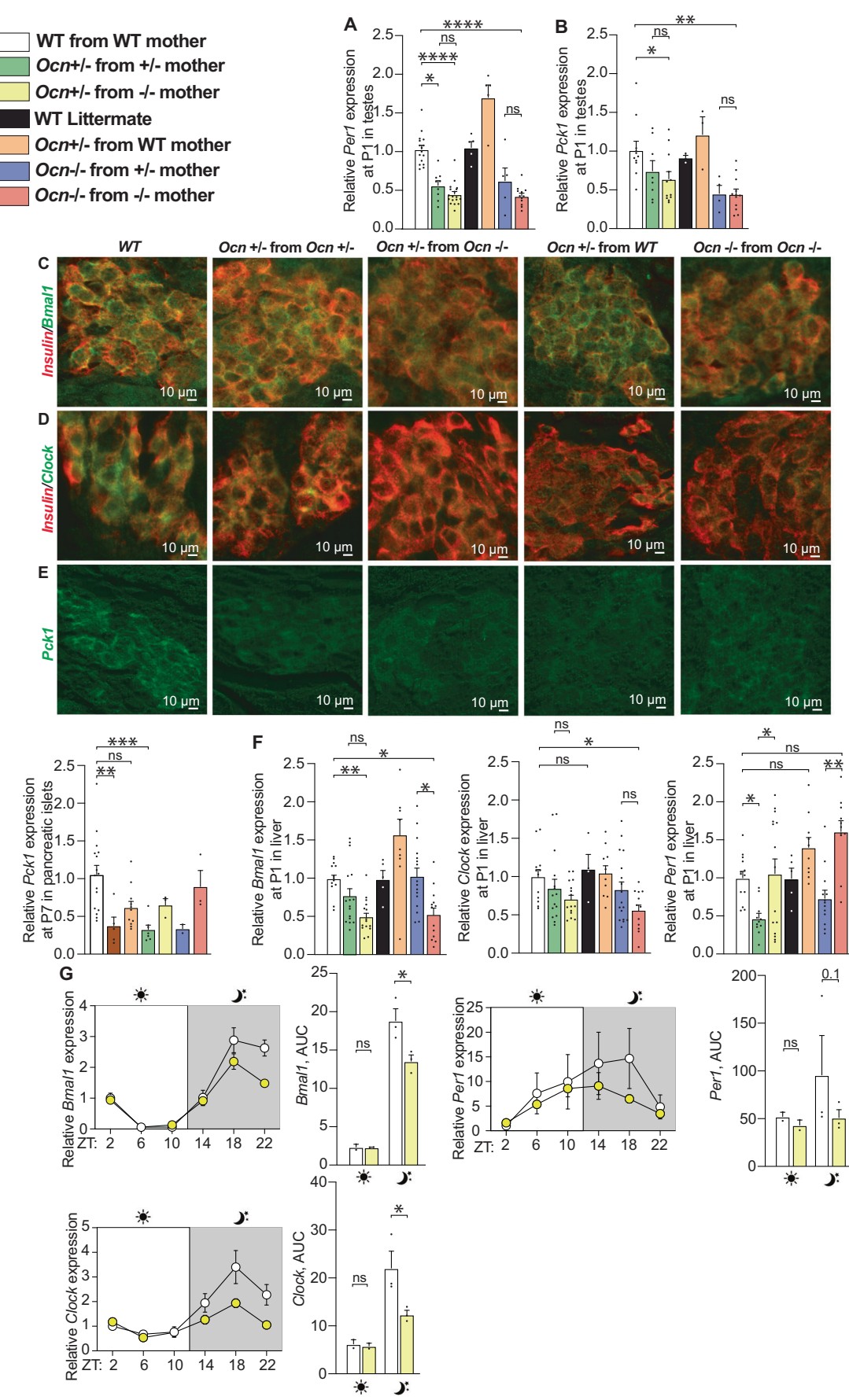

◀

**Figure 6. Expression of components of molecular clock and a gluconeogenic enzyme in the pancreas, liver and testes depends on the *Ocn* genotype of the mother.**

(A) Relative expression of circadian rhythm-related gene (*Per1*) in testes of P1 pups in different genotypes from the $Ocn_{GK}$ mouse strain. $n = 5$ or more mice per genotype analyzed. (B) Relative expression of glucose homeostasis-related gene (*Pck1*) in testes at P1 pups of different genotypes from the $Ocn_{GK}$ mouse strain. $n = 4$ or more mice per genotype analyzed. (C–E) Immunohistofluorescence analysis of Bmal1 (C), Clock (D), and Pck1 (E) levels in pancreata of P1 pups of different genotypes in the $Ocn_{GK}$ strain. In (C, D), co-immunolocalization of insulin levels was used to identify β-cells. (F) Relative expression of circadian rhythm-related gene (Bmal1, Clock, and Per1) in liver of P1 pups of different genotypes in the $Ocn_{GK}$ mouse strain. $n = 4$ or more mice per genotype analyzed. (G) Relative expressions and quantifications of area under the curve (AUC) of circadian rhythm-related genes *Bmal1, Clock, Per1* during the course of 24 h in the liver of adult mice of different genotypes in the $Ocn_{GK}$ strain. $n = 3$ or more mice per genotype analyzed. Data information: In bar plots, each dot represents an individual mouse. All data are shown as mean ± SEM. Statistical significance was determined by one-way Kruskal–Wallis test followed by post hoc multiple comparisons test. *$p < 0.05$; **$p < 0.01$; ***$p < 0.001$, ****$p < 0.0001$, ns: not significant. Source data are available online for this figure.

2012; Karsenty and Olson, 2016; Pi et al, 2021; Pi et al, 2017). This hypothesis does not exclude by any means that other hormonal and molecular events contribute to the consequences of unhealthy pregnancy on homeostasis in the offspring.

When put in an evolutionary context, the fact that haploinsufficiency for *Osteocalcin* affects several physiological processes in newborn and adult offspring that are unchallenged by a diet or any other manipulations, together with the observation that osteocalcin often acts as a regulator of regulatory molecules, whether they are hormones (insulin), cytokines (interleukin-6) or neurotransmitters (monoamines, dopamine, acetylcholine), is consistent with the notion that the appearance of bone during evolution has coincided with the acquisition of another necessary mechanism of regulation of physiology that is housed in bone (Berger et al, 2019; Lee et al, 2007; Mera et al, 2016b; Oury et al, 2011).

Another purpose of this study was to begin unraveling cellular and molecular mechanisms of osteocalcin action during gestation in multiple organs. At the cellular level, our analysis reveals that developmental osteocalcin promotes cell proliferation in pancreas and testes and thereby the size of the pools of hormones that can be released by these organs. In contrast, in neurons of the brain, it prevents cell death. In addition, and besides its own direct regulation of testicular steroidogenesis, developmental osteocalcin promotes in testes the expression of the receptor for luteinizing hormone (LH) and in that way modulates the signaling of LH in Leydig cells. This function of developmental osteocalcin in the testes is reminiscent of the function it exerts in the developing adrenal glands, where it promotes expression of the receptor of the adrenocorticotrophic hormone (Yadav et al, 2022).

To draw a roadmap of molecular signaling by developmental osteocalcin in various target organs, we performed a transcriptomic analysis in testes and forebrains of *Ocn*+/− and *Ocn*−/− newborn mice. We chose these two organs because osteocalcin signals in them through distinct receptors and they were never subjected to this type of survey. Consistent with the notion that osteocalcin exerts dominant functions, this analysis revealed that the same genes and genetic pathways were dysregulated in *Ocn*−/− and, albeit to a lesser extent, in *Ocn*+/− newborn mice in testes and other peripheral organs where osteocalcin signals through Gprc6a. Specifically, a group of genes that is consistently downregulated in testes of newborn mice lacking one or two copies of *Ocn* is the one encoding key component of the molecular clock including *Bmal1, Clock,* and *Per1*. Given that through their expression in the liver, clock genes contribute to the regulation of glucose homeostasis and energy metabolism (Perelis et al, 2015), these results suggest that osteocalcin may regulate energy metabolism in part by regulating the molecular clock in the liver. In support of this contention, we note that osteocalcin controls the circadian expression of *Pck1* in

the liver (Pi et al, 2020). A second class of genes that saw their expression downregulated in testes and the liver of *Ocn*+/− and *Ocn*−/− newborn and adult mice are those implicated in gluconeogenesis such as *Pck1*. These results highlight how central is the ability of osteocalcin to increase the intracellular availability of glucose to its functions in peripheral organs where it signals. They also distinguish the influence of osteocalcin on glucose metabolism in peripheral organs, where it is anabolic, from the one it exerts in the brain where it is favors glycolysis. We also note that in the brain and unlike what is the case in peripheral organs, in addition to genes controlling cell death, osteocalcin signaling inhibits the expression of genes involved in inflammation. Hence, osteocalcin appears to regulate different sets of genes depending on the receptor it signals through.

## Methods

### Mouse models

$Ocn_{GK}$−/− (129/SvEv) (Ducy et al, 1996), $Ocn_{BW}$−/− (Diegel et al, 2020), and $Ocn_{XL}$−/− (Qian et al, 2021) have been previously described. Wild-type (WT) mice of indicated ages and genetic backgrounds were obtained as described in Fig. 1A or purchased from Jackson laboratories or from Taconic laboratory for the analysis of $Ocn_{BW}$ mice. All animal studies were approved by Columbia University Animals Ethics Committee (Protocol AABH4550). To limit the information bias, the experimentalists were blinded to the genotype of the mice in all experiments.

### Blood glucose measurements

Random fed blood glucose levels were measured between 9:00 AM and 10:00 AM by tail blood collection. Fasting blood glucose levels were measured between 4:00 and 5:00 PM after 6 h of fasting. In both cases, glucose levels were monitored using blood glucose strips and the Accu-Check glucometer (Roche).

### Hormone measurements and dynamic metabolic tests

Blood samples were obtained at indicated ages and at indicated times mentioned below for each hormone measurement. For measurements of circulating osteocalcin, testosterone, or insulin levels, samples were collected between 9:00 AM and 9:30 AM from facial vein. Blood was collected in serum separating tubes (Microvet 500 Z Gel, Starstedt) allowed to clot for 30 min and centrifuged at 12,000 rpm for 10 min at 4 °C to obtain serum. Mouse circulating osteocalcin was measured by ELISA (Quidel kit Cat #60-1305).

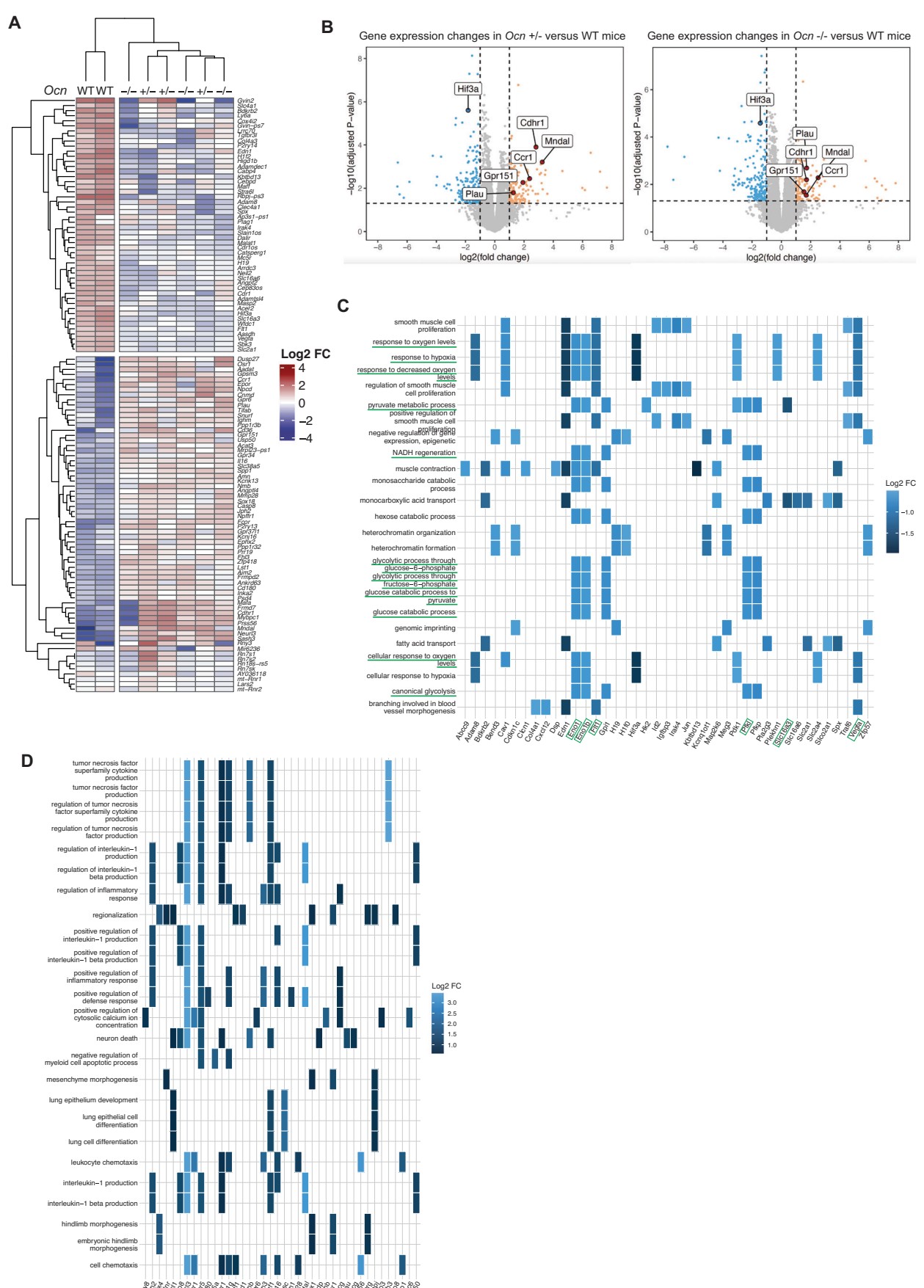

**Figure 7. Transcriptomic analysis of osteocalcin signaling in the forebrain of newborn mice of different genotypes.**

(A) Heatmap illustrating results of unsupervised hierarchical clustering of forebrains obtained from WT ($n = 2$), $Ocn+/-$ ($n = 3$) and $Ocn-/-$ ($n = 3$) mice. (B) Volcano plots of the genes differentially expressed in forebrains from $Ocn+/-$ [left panel] and $Ocn-/-$ [right panel] mutant compared to WT mice. Each dot represents one gene or pseudogene ($n = 3$). (C, D) Heatplots of the top 25 GO molecular functions (GO term) that are significantly (C) downregulated or (D) upregulated in both $Ocn+/-$ and $Ocn-/-$ compared to WT mice. Pathways and genes that are underlined in green have also been identified in the molecular signature of osteocalcin in testes.

Blood testosterone levels were determined by RIA (Testo-US Cisbio Bioassays). Blood insulin levels were measured by ELISA (Crystal Chem Cat #90080). All measurements were performed in duplicate according to the manufacturer's instructions.

Pyruvate tolerance tests were performed as described previously (Mauvais-Jarvis et al, 2002). Briefly, pyruvate (Sigma #P2256-25G) was injected intraperitoneally (IP) (2.5 g/kg body weight (BW)) after 6 h of fast. Blood glucose was monitored using blood glucose strips, and the Accu-Check glucometer (Roche) at indicated times (Lee et al, 2007). Glucose stimulated insulin tolerance tests (GSIS) were performed as described previously (Lee et al, 2007). Briefly, glucose (3 g/kg BW) was injected IP after an overnight fast. Blood was collected at indicated times from the tail in serum separating tubes (Microvet 100 Z, Starstedt), allowed to clot for 30 min, and centrifuged at 12,000 rpm for 10 min at 4 °C to obtain serum (Lee et al, 2007). After that, insulin levels were measured by ELISA (Crystal Chem Cat #90080).

## Behavioral analyses

For all behavioral assessments, mice were handled for at least 20 min over 3 consecutive days. Before the behavioral procedure, mice were maintained in the testing room, for at least 1 h in their home cage prior to testing. Behavior was scored by two observers blind to the groups.

### Novel object recognition paradigm (NOR)

The testing arena consisted of a plastic box ($60 \times 40 \times 40$ cm). Mice could not see each other during the experiment. Two different objects were used: (A) a black ceramic pot (diameter 6.5 cm, maximal height 7.5 cm) and (B) a clear/plastic funnel (diameter 8.5 cm, maximal height 8.5 cm). The objects elicited equal levels of exploration as determined in control experiments and training phases. Mice were placed in the center of the arena at the start of each exposure. Sessions were recorded with a video camera. The NOR paradigm consisted of three phases over 2 days. On day 1 (habituation phase), mice were given 5 min to explore the arena devoid of objects and were then taken back to their home cage. On day 2, mice were first allowed to explore for 5 min two identical objects arranged in a symmetric opposite position from the center of the arena and were then transported to their home cage. One hour later (testing phase), mice were given 5 min to explore two objects: a familiar object and a novel one, in the same arena, keeping the same object location. Between exposures, arenas were cleaned with a disinfectant (Phagospore), and the bedding was replaced. The following behaviors were considered as exploration of the objects: sniffing or touching the object with the nose or with the front legs or directing the nose to the object at ≤1 cm distance. Exploration was not scored if the mouse was on top of the object or completely immobile. Total exploration time was quantified during the training and testing phases. The preference index (time spent exploring the new object/the total time spent exploring both objects) and the discrimination index (time spent exploring the new object − time spent exploring the familiar object)/(total time spent exploring both objects) were calculated. As a control, the preference index for the (right/left) object location or for the object A versus B during the training phase of the NOR was measured.

## RNA sequencing

Total RNA was isolated from fresh forebrains and testes obtained from one-day-old male mice of the following genotypes: $Ocn_{GK}$ WT born from WT mothers, $Ocn_{GK}-/-$ born from $Ocn_{GK}-/-$ mothers, and $Ocn_{GK}+/-$ born from $Ocn_{GK}-/-$ mothers. All Testes were collected at same time of the day (between 6 and 7 pm) RNA extraction was performed using the QIAGEN RNeasy isolation kit (Ref#74104) following the manufacturer's instructions, followed by NanoDrop RNA quantification (Thermo Fisher Scientific). The quality of purified RNA samples was determined by Bioanalyzer 2100 (Agilent) using the RNA Nano kit (Berger et al, 2019). RNA Integrity Number (RIN) was verified for each RNA sample and only samples with a RIN score of 8–10 were only utilized for cDNA synthesis and library preparation. RNA sequencing was performed by the Columbia University Genome Centre. Briefly, following the manufacturer's manual, sequencing libraries were generated using NEBNext Ultra™ RNA Library Prep Kit for Illumina (NEB, USA). In all, 2–3 µg RNA per sample was used to purify mRNA using poly-T oligo attached magnetic beads and then the purified mRNA will be fragmented into short fragments (about 200 bp) by the fragmentation buffer. First-strand cDNA was synthesized using random hexamer primer and Reverse Transcriptase. NEBNext Adapter were ligated after the adenylation of 3' ends of DNA fragments. Complimentary DNA fragments 150–200 bp in length were selected for PCR amplification to create cDNA libraries. Libraries were sequenced on an Illumina Hiseq 2500 platform to generate the sequences.

## Bulk RNA-seq data analysis

Quality checks were performed with FastQC version 0.11.9 and processed with Trimgalore version 0.6.6 to exclude low-quality bases. Sequence reads were aligned with STAR version 2.7.8a and calculated read counts with RSEM version 1.3.3 against the mouse reference sequence (GRCm39). Calculating counts per million (CPM) and differentially expressed genes were performed using edgeR package version 3.38.4 in R version 4.2.2. Heatmaps were drawn by using the pheatmap function from the pheatmap package version 1.0.12 in RStudio version 2023.06.0+421. Volcano plots were drawn using the ggplot function from ggplot2 package version 3.4.2 in RStudio. Category netplots and heatplots were respectively drawn using the cnetplot and heatplot functions from enrichplot package version 1.20.0 in RStudio.

## Gene expression analysis

All tissues were collected between 6 and 7 pm for all mice of all genotypes and all strains. Total RNA was isolated from the testes, liver, or forebrain using a Qiagen RNA isolation kit (Ref#74104). RNAs were quantified with nanodrop and reverse transcription performed with 1 μg RNA in 20 μl volume. One μl of cDNA was used for qRT-PCR analysis with SYBR green method (Applied Biosystems) of respective genes and 18s ribosomal RNA was used as an internal control. cDNA for the internal control was diluted 50–500× to reach CT value within 5–6 cycles of the gene whose expression was being tested. qRT-PCR end products were run on a 2% agarose gels to confirm specificity of the primers. Gene expression was reanalyzed with standard qRT-PCR in the linear range of amplification and run on a 2% agarose gel to confirm the change observed through the RT-qPCR.

qPCR primer sequences used for SYBR Green-based qPCR assays were as follows: *Pck1F*: AGAACAAGGAGTGGAGACCG; *Pck1R*: TCCTACAAACACCCCATGCT; *G6pcF*: CTGTCTGTCCCGGATCT ACC; *G6pcR*: GCGCGAAACCAAACAAGAAG; *StarF*: AAGAGCTC AACTGGAGAGCAC; *StarR*: TACTTAGCACTTCGTCCCCGT; *Cyp11a1F*: AGGTCCTTCAATGAGATCCCTT; *Cyp11a1R*: TCCCTG TAAATGGGGCCATAC; *18sF*: CGCGGTTCTATTTTGTTGGT; *18sR*: AGTCGGCATCGTTTATGGTC; *Bmal1F*: TCCTCAACCATCA GCGACTT; *Bmal1R*: TTCAATCTGACTGTGGGCCT; *ClockF*: ATGC CACAGAACAGTACCCA; *ClockR*: TTGTGTGGCGAAGGTAGGAT; *Cry1F*: CAGAGGGCTAGGTCTTCTCG; *Cry1R*: GTCCCCGTGAGC ATAGTGTA; *Per1F*: AATGGCAAGGACTCAGCTCT; *Per1R*: CGAA GTTTGAGCTCCCGAAG; *PyglF*: TACATTCAGGCTGTGCTGGA; *PyglR*: AAGGCATCAAACACGGTTCC; *LhcgrF*: ACCCGGTGCTTT TACAAACC; *LhcgrR*: CGTCGTCCCATTGAATGCAT.

## Histological and histomorphometric analysis

P1 pups were intraperitoneally injected with BrdU (100 mg/kg; Sigma-Aldrich). Their pancreata were fixed in 4% PFA in PBS for 12 h at 4 °C followed by dehydration, paraffin embedding, and histologically analyzed as described previously (Lee et al, 2007; Wei et al, 2014). For β-cell proliferation analysis in P1 pups every other section was labeled with guinea pig anti-insulin (1:200; DAKO A0564) and mouse anti-BrdU (1:500 Sigma B2531) antibodies, followed by fluorochrome-coupled secondary antibodies (Millipore) and DAPI counterstaining (Abcam). β-cell proliferation was quantified by counting the number of BrdU/insulin-positive cells over the total number of insulin-positive cells using ImageJ software. An average of 2000 insulin-positive cells per specimen was counted. Bmal1 and Clock levels were assessed on pancreas sections of P1 pups following co-labeling with rabbit anti-Bmal1 (1:500 Abcam ab3350) or rabbit anti-Clock (1:1000 Abcam ab3517) antibodies and a guinea Pig anti-insulin (1:800 DAKO A0564) antibody while Pck1 levels were assessed using a rabbit anti-PCK1 (1:400 Abcam ab70358) antibody, followed by fluorochrome-coupled secondary antibodies (Invitrogen A21206 and A21450).

## Whole genome sequencing analysis of CRISPR targeted *Ocn*<sub>BW</sub>−/− and *Ocn*<sub>BW</sub>+/+ mice

### Library preparation and sequencing

Long read whole-genome sequencing libraries targeting N50:20 kb reads were prepared using the Ligation Sequencing Kit (Oxford Nanopore Technologies, LSK-110) with 10 μg of DNA input to achieve N50:10 kb median read length. Library preparation followed Oxford Nanopore's recommended procedures, with minor modifications. Samples were aliquoted from the Matrix rack tubes and fragmented using a g-TUBE (Covaris, 520104). After shearing samples were reviewed on the Agilent TapeStation to confirm the desired range, a bead-based cleanup was performed to remove small fragments, followed by another quality control. This was followed by end-prep and nick repair, and then by ligation of sequencing adapters. Final libraries were quantified by Qubit (ThermoFisher, Q32854) prior to being loaded on a 9.4.1 Flow Cell on the Oxford Nanopore Technologies PromethION, one sample per Flow Cell. Nuclease flushes were performed on the PromethION at 24-48 h.

### Base calling, assembly and variant calling

The raw sequencing signal data (in FAST5 format) from the Promethion P24 sequencer were converted into DNA base calls (in FASTQ format) using Oxford Nanopore's Guppy basecaller v6.1.2 in super high accuracy (SUP) mode. The base called sequencing reads were de novo assembled using Flye v2.9-b1768 with default (Lin et al, 2016).

For comparing WT *Ocn*<sub>BW</sub> and *Ocn*<sub>BW</sub>−/− against WT 129/ SvEv the flye assemblies for WT *Ocn*<sub>BW</sub> and *Ocn*<sub>BW</sub>−/− were aligned to the WT 129/SvEv assembly using minimap2 v2.24-r1122 (Li, 2018, 2021). The alignments were then used to call 1. Small variants i.e., single nucleotide polymorphisms (SNPs), insertions and deletions (InDels) less than 50 bp and 2. Large variants, i.e., structural variants (SVs) greater than 50 bp using the minimap2's "paftools.js call" tool.

## In situ hybridization and BrdU labeling analyses in testes

Testes of adult mice were fixed in 4% PFA in PBS for 12 h at 4 °C followed by dehydration and paraffin embedding. Five-7μm sections were used for in situ hybridization analysis using ACD RNAscope kits (RNAscope® Multiplex Fluorescent Reagent Kit v2). *Sf1* probe was obtained from ACD RNAscope (catalog # 445731). Serial sections were used for detection of BrdU labeled interstitial cells. All images were taken with identical laser settings and pseudo-color coding in the images was done for better comparative visualization.

## Statistics

All analyses described were performed with RStudio v1.2.5033 (RStudio Team (2019), RStudio: Integrated Development for R, RStudio, Inc., Boston, MA, URL http://www.rstudio.com/). For all measurements, samples size was <30 and contained outliers, thus, we could not assume neither the Gaussian distribution of the variables, nor the homoscedasticity of the variances. Consequently, non-parametric tests have been chosen for hypothesis testing. Comparisons of means between two groups were performed by one-sided Mann–Whitney testing with the R function Wilcoxon test from *stats* package version 4.1.2. For GSIS experiment, one-sided paired Wilcoxon signed-rank test has been used to compare insulin secretion at the baseline and two minutes after glucose injection. To do so, the R function *Wilcoxon test* from *stats* package version 4.1.2. has been used. Comparisons of means between more

than two groups were done by Kruskal–Wallis testing followed by post hoc Dunn's multiple comparisons test with the function *dunnTest* from *FSA* package version 0.9.3. *p* value lower than 0.05 was considered significant. Bar plots were drawn with GraphPad Prism 9. Data are presented as means ± standard error of the mean (SEM). In all figures, statistical significance is indicated by stars $*p < 0.05$, $**p < 0.01$, $***p < 0.001$, $****p < 0.0001$. An absence of star or the presence of "ns" indicates the absence of significant difference.

## Data availability

Mouse strains are all publicly available. RNAseq data are available at https://www.ncbi.nlm.nih.gov/geo/query/acc.cgi?acc=GSE248211 and the GSE number is GSE248211. The whole-genome sequencing data are available at http://www.ncbi.nlm.nih.gov/bioproject/1044076 and the Bioproject ID is PRJNA1044076. The data generated in this study are available in this article and in Source data files.

## Peer review information

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

## Acknowledgements

We thank Dr. Hong Liu for expert technical assistance. Long-read whole-genome sequencing (Oxford Nanopore) and bioinformatics analysis (variant calling and whole-genome assembly) were performed at the New York Genome Center. Histomorphometric analyses were performed by the Histology and Histomorphometry Core. This work was supported by NIH grants P01 AG 032959-13, R01 DE 027887-05, R01 HD 107574-02, and R01 AR 073180-04. JMB was supported by a Druckenmiller fellowship from the NY Stem Cell Foundation. Dr. Xiang Li was supported by a Shenzhen Science and Technology Program (Project number: KQTD20210811090117032).

## Author contributions

**Danilo Correa Pinto Junior**: Investigation; Writing—original draft; Writing—review and editing. **Isabella Canal Delgado**: Investigation; Writing—original draft; Writing—review and editing. **Haiyang Yang**: Investigation; Writing—original draft; Writing—review and editing. **Alisson Clemenceau**: Investigation; Writing—original draft; Writing—review and editing. **André Corvelo**: Data curation; Formal analysis; Methodology; Writing—original draft. **Giuseppe Narzisi**: Data curation; Formal analysis; Methodology; Writing—original draft. **Rajeeva Musunuri**: Formal analysis; Methodology. **Julian Meyer Berger**: Investigation; Writing—original draft; Writing—review and editing. **Lauren E Hendricks**: Investigation. **Kazuya Tokumura**: Formal analysis; Investigation. **Na Luo**: Investigation. **Hongchao Li**: Investigation. **Franck Oury**: Formal analysis; Supervision; Methodology; Writing—original draft; Project administration; Writing—review and editing. **Patricia Ducy**: Supervision; Funding acquisition; Writing—original draft; Project administration; Writing—review and editing. **Vijay K Yadav**: Funding acquisition; Writing—original draft; Writing—review and editing. **Xiang Li**: Data curation; Writing—original draft; Project administration; Writing—review and editing. **Gerard Karsenty**: Conceptualization; Supervision; Funding acquisition; Writing—original draft; Project administration; Writing—review and editing.

## Disclosure and competing interests statement

The authors declare no competing interests.

# Expanded View Figures

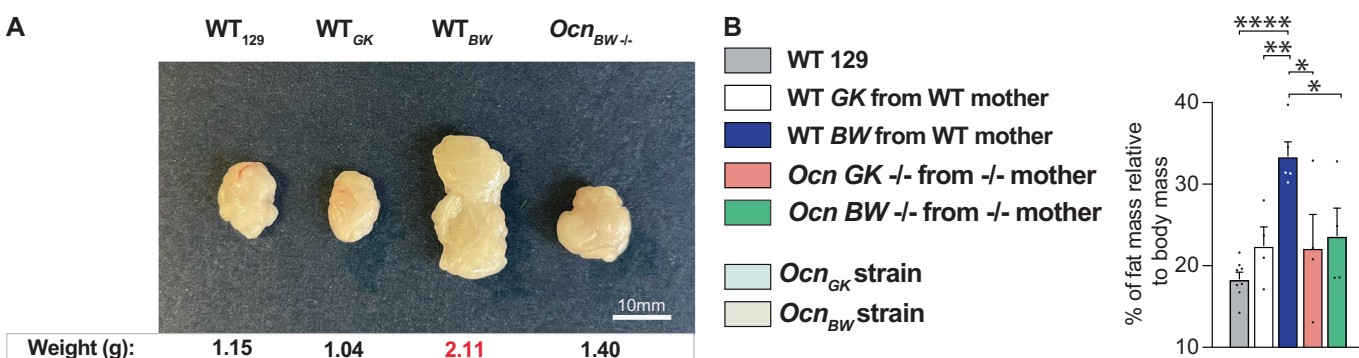

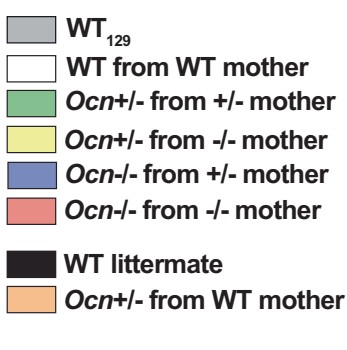

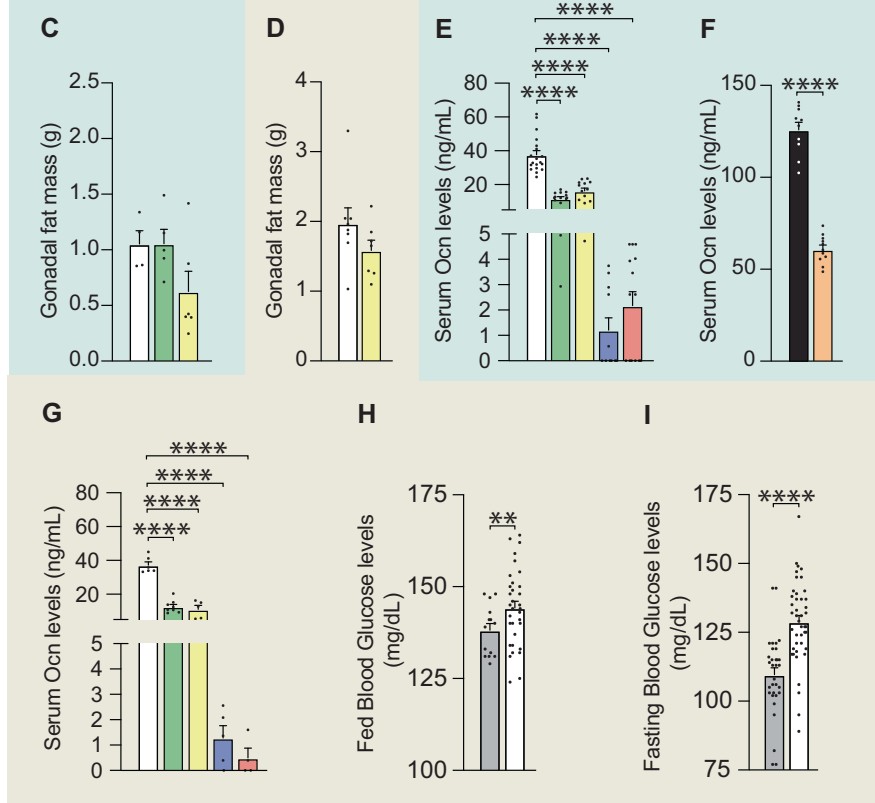

**Figure EV1. Characterization of three mouse models of *Ocn* deletion.**

(A) Photomicrographs of gonadal fat pads from $WT_{129}$, $WT_{BW}$, and $Ocn_{BW}-/-$ mice. Weight for each group is shown below each representative fat pad image. (B) Percent fat mass relative to body weight in $WT_{129}$, $WT_{BW}$, and $Ocn_{BW}-/-$ mice assessed by EchoMRI. $n = 4$ or more mice per genotype analyzed. (C, D) Gonadal fat mass in different genotypes of 12 weeks-old $Ocn_{GK}$ mice (C). $n = 4$ or more mice per genotype analyzed. Gonadal fat mass in different genotypes of 12 weeks-old $Ocn_{BW}$ mice (D). $n = 7$ mice per genotype analyzed. (E–G) Serum Osteocalcin levels in different genotypes of 12 weeks-old (E) and 5 weeks-old (F) $Ocn_{GK}$ mice. $n = 11$ or more mice per genotype analyzed. Osteocalcin levels in different genotypes of 12 weeks-old $Ocn_{BW}$ mice (G). $n = 4$ or more mice per genotype analyzed. (H, I) Fed (H) and fasting (I) glucose levels in $Ocn_{BW}$ WT and WT129 mice. $n = 14$ or more mice per genotype analyzed. In bar plots, each dot represents an individual mouse. All data are shown as mean ± SEM. Statistical significance was determined by one-way Kruskal–Wallis test followed by post hoc multiple comparisons test. $*p < 0.05$; $**p < 0.01$, $****p < 0.0001$. Source data are available online for this figure.

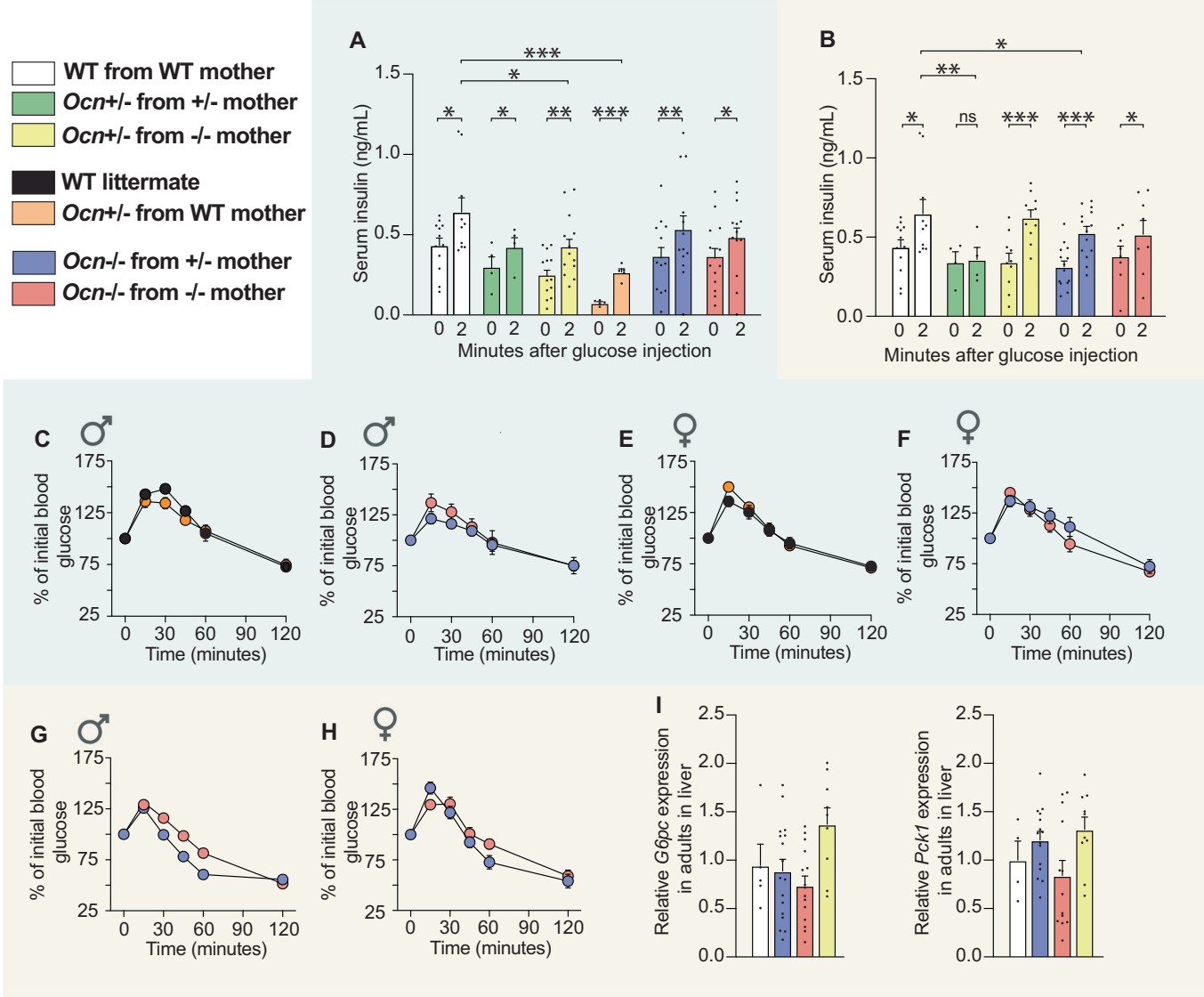

**Figure EV2. Dynamic analysis of glucose metabolism in *Ocn-deficient* mouse strains.**

(A, B) Serum insulin levels at 0 and 2 min after glucose injection in glucose stimulated insulin secretion test in mice of indicated genotypes of $Ocn_{GK}$ mouse strain (A). $n = 4$ or more mice per genotype analyzed. Serum insulin levels at 0 and 2 min after glucose injection in glucose stimulated insulin secretion test in mice of indicated genotypes of $Ocn_{BW}$ mouse strain (B). $n = 7$ or more mice per genotype analyzed. (C–H) Serum glucose levels at different time points post injection of pyruvate in a pyruvate tolerance test in mice of indicated genotypes in the $Ocn_{GK}$ (C–F) mouse strain. Serum glucose levels at different time points post injection of pyruvate in a pyruvate tolerance test in mice of indicated genotypes in $Ocn_{BW}$ (G, H) mouse strain. $n = 6$ or more mice per genotype analyzed. (I) Relative Pck1 and G6pc expression in the liver from adult mice of different genotypes in the $Ocn_{BW}$ strain. $n = 4$ or more mice per genotype analyzed. In bar plots, each dot represents an individual mouse. All data are shown as mean ± SEM. Statistical significance was determined by one-way Kruskal–Wallis test followed by post hoc multiple comparisons test. *$p < 0.05$; **$p < 0.01$; ***$p < 0.001$ ns: not significant. Source data are available online for this figure.

**A**

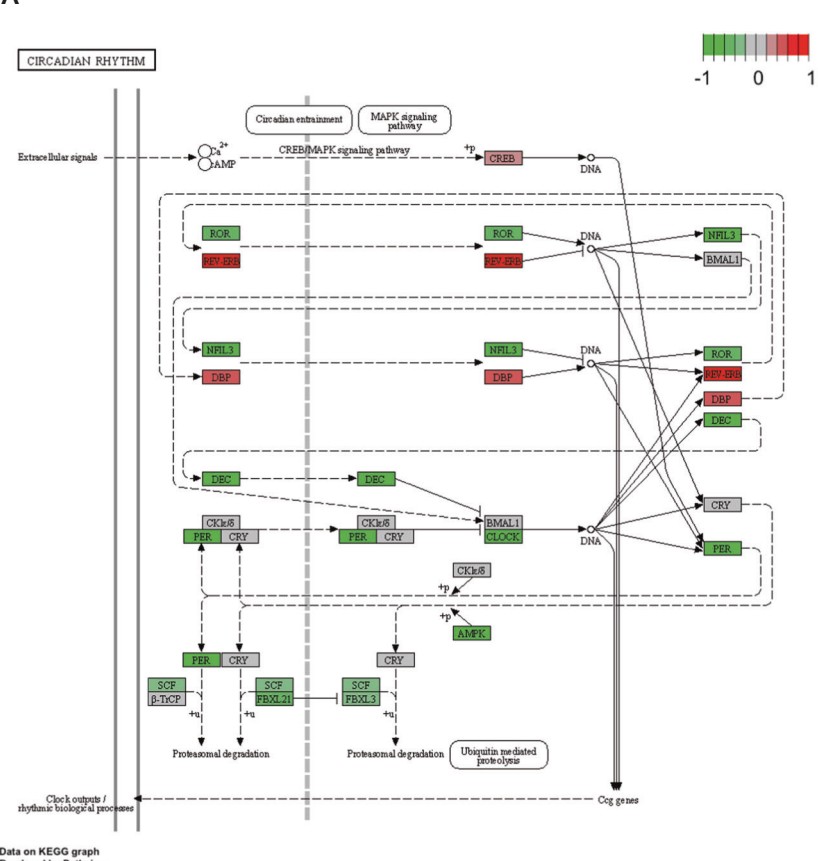

**B**

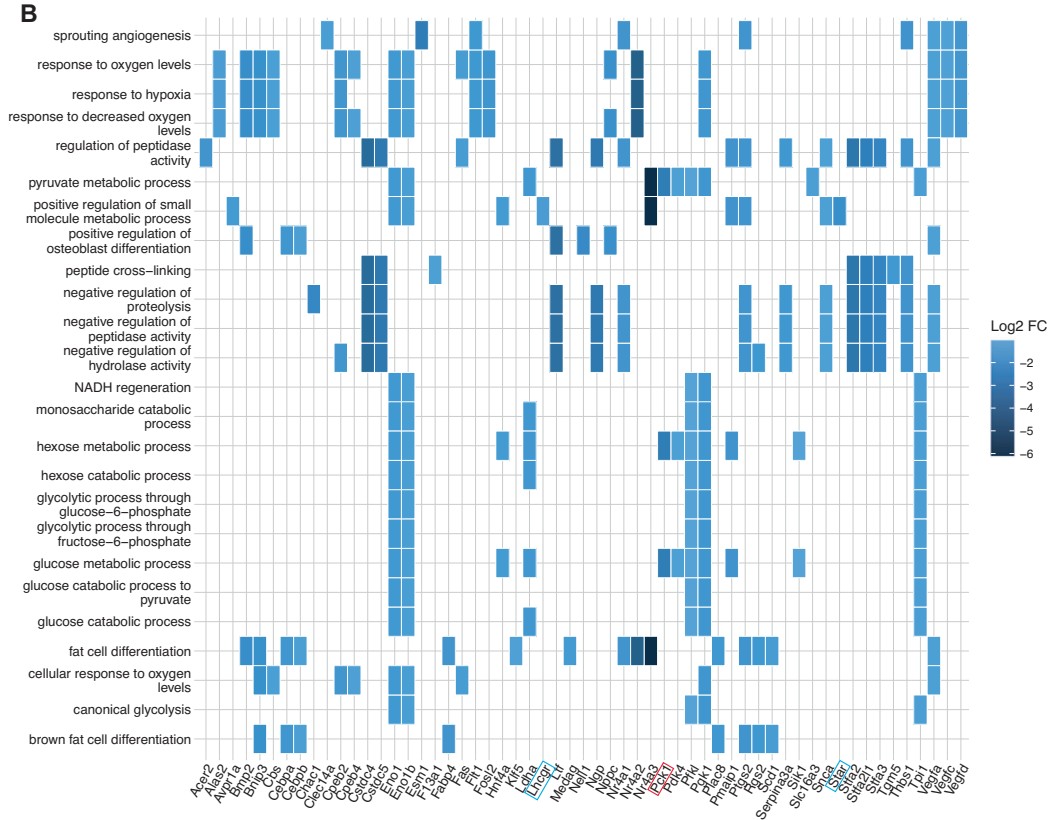

◄   **Figure EV3.   KEGG analysis of transcriptomic studies of testes gene expression.**

(A) KEGG pathview graph of the molecular clock comparing the transcriptomic data obtained from *Ocn−/−* mice to those from WT mice. Values close to −1 (green) indicate a downregulation of the gene in *Ocn−/−* compared to WT, values close to 0 (gray) indicate no difference and values close to 1 (red) indicate an upregulation in *Ocn−/−* compared to WT. (B) Heatplots of the top 25 GO biological processes (GO term) that are significantly downregulated in both *Ocn+/−* and *Ocn−/−* compared to WT mice. *n* = 2-3 mice per genotype analyzed.

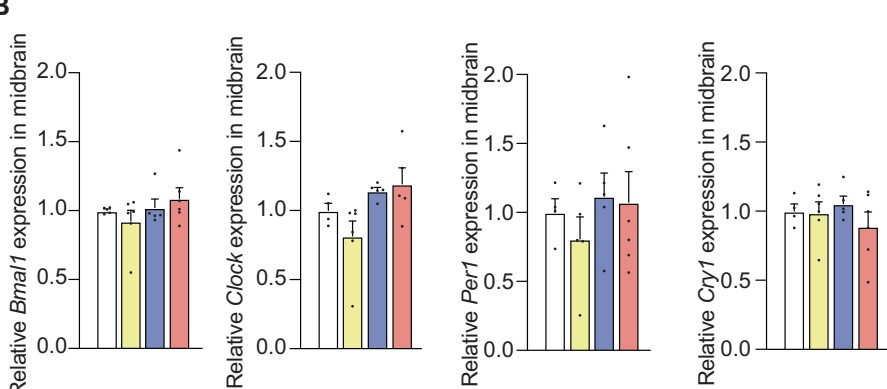

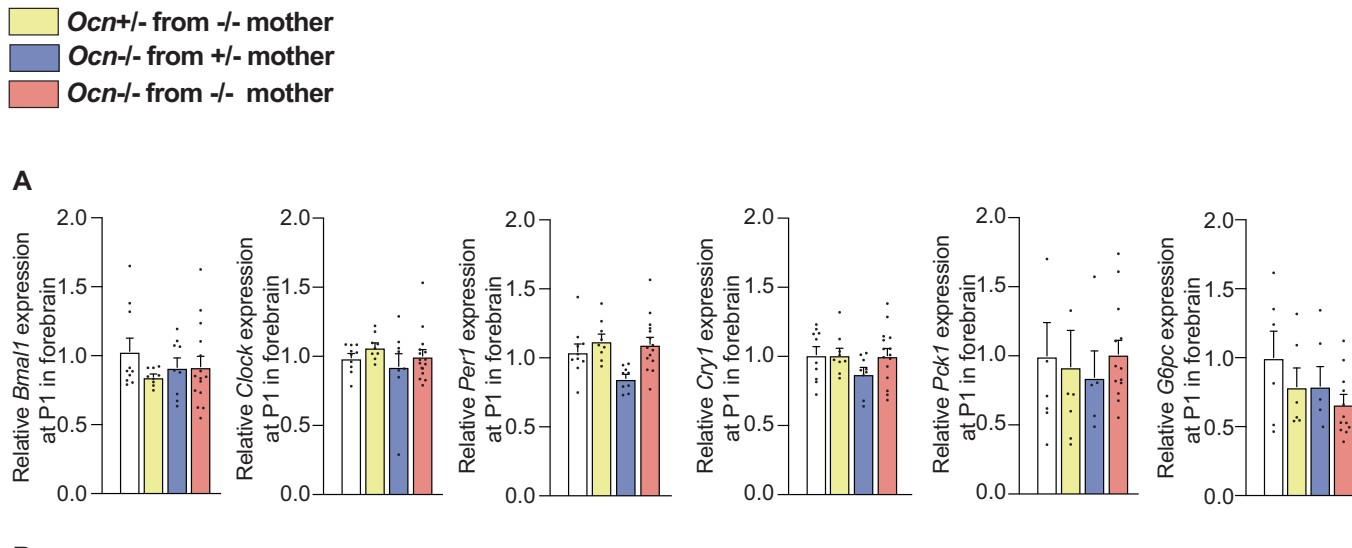

**Figure EV4. Gene expression analysis in forebrain (newborn) and midbrain (adults).**

(A) Relative expressions of circadian rhythm-related genes, Bmal1, Clock, Per1, and Cry1, and glucose homeostasis genes, Pck1 and G6pc in forebrain at P1 in different genotypes from Ocn$_{GK}$ mouse strain. $n = 9$ or more mice per genotype analyzed. (B) Relative expressions of circadian rhythm-related genes Bmal1, Clock, Per1 and Cry1 in midbrain of adult mice of different genotypes from Ocn$_{GK}$ mouse strain. $n = 4$ or more mice per genotype analyzed. In bar plots, each dot represents an individual mouse. All data are shown as mean ± SEM. Statistical significance was determined by one-way Kruskal–Wallis test followed by post hoc multiple comparisons test. Source data are available online for this figure.

