## [Peer Review File · EMBO Reports]

Osteocalcin of maternal and embryonic origins synergize to establish homeostasis in offspring

Danilo Correa Pinto Junior, Isabella Canal Delgado, Haiyang Yang, Alisson Clemenceau, André Corvelo, Giuseppe Narzisi, Rajeeva Musunuri, Julian Meyer Berger, Lauren Hendricks, Kazuya Tokumura, Na Luo, Hongchao Li, Franck OURY, Patricia Ducey, Vijay Yadav, Xiang Li, and Gerard Karsenty

DOI: [10.15252/embr.202358057](https://doi.org/10.15252/embr.202358057)

Corresponding author(s): Gerard Karsenty (gk2172@cumc.columbia.edu), Franck OURY (franck.oury@inserm.fr), Patricia Ducey (pd2193@cumc.columbia.edu), Vijay Yadav (vky2101@cumc.columbia.edu), Xiang Li (xiang.li@siat.ac.cn)

Review Timeline:

Submission Date:	25th Aug 23
Editorial Decision:	14th Nov 23
Revision Received:	14th Nov 23
Editorial Decision:	23rd Nov 23
Revision Received:	24th Nov 23
Accepted:	1st Dec 23

Transaction Report:

(Note: With the exception of the correction of typographical or spelling errors that could be a source of ambiguity, letters and reports are not edited. Please note that the manuscript was transferred from another journal where it was originally reviewed. Since the original reviews are not subject to EMBO's transparent review process policy, they cannot be published.)

At the onset of this rebuttal letter, we want to thank all reviewers for their constructive criticisms. We have addressed all of them; most of them experimentally, and others by clarifying the context of these discoveries in the field. Addressing these criticisms, have I believe, improved the quality and clarity of this paper. Please find below a point-by-point answer to each comment of each reviewer.

Reviewer #1

“1) In figure 1 C-F it does look like the paternal genotype has an impact on blood glucose level, since the WT from WT father have higher fed glucose than the WT with +/- father. This is also the case for the fasted glucose of the OCN BW in Fig. 1F. Moreover, insulin (Fig. 2A) is lower in WT from WT father than in WT with +/- father. These interesting observations are not discussed and may indicate some epigenetic effect of paternal osteocalcin on the offspring. This is plausible since osteocalcin has an important function in the testis.”

The reviewer raises here an important point that we had not thoroughly addressed previously. In these figures, WT mice born from WT or *Ocn*^{+/-} fathers were significantly younger than they were in other groups. We have now performed these measurements in mice of the same age (3-month-old) and the differences noted by us and the reviewer have disappeared, so we do not think that there is a strong paternal epigenetic contribution to the phenotypes.

2) Fat is another target tissue of osteocalcin in adult. Does maternal and embryonic osteocalcin influence fat mass in the offspring? This should be at least discussed.”

We now show that the inguinal fat pad weight is similar in adult WT mice born from WT mothers, and *Ocn*^{+/-} mice born from *Ocn*^{+/-} or *Ocn*^{-/-} mothers. These data are now presented in Supplemental Figure 1 Panel C and D, and mentioned in the text page 8, Lines 1 to 4.

3) For the OCN XL strain, some of the groups analyzed in the other strains are missing. Could the authors explain why?”

The group of Dr. Li, which generated the OCN_{XL} strain, specializes in neuroscience (Qian *et al*, 2021), and preferred not to embark in any dynamic metabolic tests for which they have no training. This is mentioned in the text page 6, Line 16.

“4) Pyruvate tolerance test are normally displayed in absolute values (mg/dl or mmol/L of glucose). Could the authors provide the rational for presenting their data in percentage of initial glucose?”

We have used the classical representation of PTT recommended by the American Diabetes Association. We could, however, represent PTT in absolute value if requested by the Editor.

“Similarly, it is not clear why testosterone is shown as a percentage in Fig 3A.”

To comply with this request from the reviewer, we now present the absolute values for testosterone measurements (Figure 4A-B).

“5) In figure 3 and 5, the group labeled as “WT littermate from WT mother” (black bars) is missing. Were these mice different from WT from WT mother (white bar)? In this case, this may again indicate some paternal impact of the Ocn genotype (see point 1).”

We apologize for the error. We have now included these data in Figure 3 panel J, and 6 panel A, B, and F.

“6) Could the histological data in figure 5C-E quantified on multiple animals to get statistically meaningful results?”

These experiments were done by immunofluorescence that does not lend itself to statistical analyses. To address this concern of the reviewer we could, if requested by the Editor, isolate pancreatic islets and perform gene expression analysis. This would require a couple of weeks.

“7) The description of the results of the whole genome sequencing of the WT mice from the Ocn BW strain is a bit confusing. Was the genome of the Ocn BW -/- also sequenced as an appropriate comparison? Are the 11,000 insertions and deletions also present in the Ocn BW -/- mice? Conceivably, those variations are coming from the C3H background. Moreover, if the WT mice on this strain are obese, why the Ocn-/- are normal? This may suggest that the absence of Ocn prevent obesity in this background due to reduced circulating insulin, as was previously shown in other models by the group of Jim Johnson for instance. In other word, the lean phenotype of the OCN KO on this obese background may be explained by some of the known osteocalcin biological functions. A second possibility could be that the obesogenic variant(s) is/are close to the Ocn locus and genetically linked to the WT allele. This could be easily verified if both the WT and KO mice were sequenced. In conclusion, this part deserves some clarification.”

We apologize to the reviewer for a lack of clarity on our part. We did perform whole-exome sequencing in WT C57 and WT C3H, and in *Ocn_{BW}* -/- null mice. We had previously sequenced the entire *Osteocalcin* locus in 129Sv/Ev mice (Desbois *et al*, 1994). We now present a schematic representation of our findings. We also specify that within the *Osteocalcin* locus, there were mutations present in the WT *Ocn_{BW}* mice that were NOT present in WT 129, C57 or C3H reference genomes. These mutations were also not present in *Ocn_{BW}* -/- mice either.

We believe the reviewer is right, the absence of obesity in *Ocn_{BW}* -/- mice can be partly due to their reduced circulating insulin levels, and we have now inserted this statement on page 6, lines 2-11. However, given the extent of rearrangements of the *Osteocalcin* locus in *Ocn_{BW}* WT mice, we are reluctant to state that this is the only reason for the absence of obesity in *Ocn_{BW}* -/- mice.

“8) For the RNAseq and qPCR studies, were the tissue collected at the same time of the day for all the newborns? This is important since clock genes were found to be downregulated in the Ocn+/- and -/- mice and these genes oscillate over a 24h period.”

We now specify in Materials and Methods that for gene expression experiments tissues were collected between 6 and 7 pm for ALL the newborns analyzed.

“9) Page 4 : « ...using as controls WT and Ocn-/- mice born from either Ocn+/- or Ocn-/- mothers. ». This sentence seems to imply that some controls are WT (+/+) born from Ocn-/- mother, which is genetically impossible, unless embryo were transferred into pseudo pregnant mothers.”

We apologize for this poor description, we have now clarified the text.

“10) Page 5: “In each strain, the absence of circulating osteocalcin in Ocn-/- mice was verified (Figure S1D-F).” Although the levels are indeed very low in the Ocn-/- born from +/- or -/- mother, they are not null, which may be surprising since both osteocalcin genes have been inactivated in these mice. Is this due to non-specific signal of the ELISA assay used here or to the expression of the third osteocalcin gene (Blap3/ORG)?”

We had shown earlier that the third gene is not transcribed (Tabernero *et al*, 1997). These residual levels of osteocalcin are non-specific signal of the ELISA we used.

11) Page 10: “To begin addressing this question, we performed an RNA sequencing (RNA-seq) analysis in testes of newborn WT, Ocn+/- and Ocn-/- mice born from Ocn-/- mothers.” And page 14: “Here again, this analysis was performed in newborn WT, Ocn+/- and Ocn-/- mice born from Ocn-/- mothers.” These sentences are not clear: WT cannot be born from Ocn-/- mothers. Does it mean that the WT were born from WT or heterozygous mothers, and the two other groups from Ocn-/- mothers?

We apologize for a lack of clarity on our part. The WT mice were born from the WT mothers and the two other groups (Ocn+/- and Ocn-/-) were from Ocn-/- mothers. We now make this point in the paper on page 12, Lines 9 and 10.

12) There is a glitch in Fig. 2E where two bars of different color are overlapping.

We have corrected this error.

13) The quantification in Fig. 3C is incorrectly labeled “% of decrease”. It should be “% of BrdU positive”.

We have corrected this error.

Reviewer #2:

1) To understand the maternal effect of osteocalcin, it would be helpful by tracking the osteocalcin level in placenta, embryo and mother during the developmental stage.

The reviewer raises here an important point. We have previously measured circulating osteocalcin levels in WT and *Ocn*^{-/-} embryos at various ages and had reported these data (Oury *et al*, 2013). This is why we did not present them anew in this manuscript. We now refer to this figure in the text, page 5, lines 2-5. We have also measured circulating osteocalcin levels in pregnant females. The data are presented in a figure **for the reviewer perusal**. However, we prefer not to publish this latter data as they are part of a manuscript being prepared by one of the co-authors (Dr. P. Ducy).

*2) Regarding *Ocn*^{BW} mouse strain, its wt strain already displayed obese phenotype. It lacks rationale to use WT129/SvEv mice as controls. The rationale provided in this study is confusing. How could the authors estimate the difference between the offspring glucose homeostasis coming from osteocalcin or simply because of other issues of this particular genetic strain?*

We apologize for a lack of clarity on our part and have addressed this concern in full. Since *Ocn*^{BW} mice were backcrossed 4 times to 129Sv/Ev mice, their genome is over 70% 129 according to standard genetic background crosses (Benavides *et al*, 2020). This is why we consider *Ocn*^{BW} mice to be mostly albeit, not only 129. To further address this concern of the reviewer, we now have included in Figures 2 and 3, littermate controls of *Ocn*^{BW} mice when they were tested. This does not change the results. For testes analysis, we always used WT *Ocn*^{BW} mice as controls, this is mentioned in the text page 6, line 11.

*3. In Figure 2 J and K, relative *Pck1*, *G6pc*, and *Pygl* expression in the liver from P1 pups of from *OCN*^{-/-} mother (yellow bars) is dramatically decreased comparing to the pups from *OCN*^{+/-} mother (green bar); however, the decreases seem contradictory with the Figure 2 F to I. Why? In addition, what about these changes in adult off springs?*

We did not report the expression of these genes in *Ocn*^{-/-} mice since these data have already been presented for mice lacking the osteocalcin receptor in hepatocytes (Pi *et al*, 2020), and we did not want to submit results already published. However, we measured expression of these three genes in adult *Ocn*^{+/-} mice of each strain, and failed to observe a difference between WT and *Ocn*^{+/-} mice. These data suggest that the decrease in liver gluconeogenesis one can measure by PTT in adult *Ocn*^{+/-} mice is at least in part of developmental origin. These data are now presented in supplemental Figure 3, panel I and mentioned in the text page 10, lines 11 to 17.

4. This study evaluated functional processes in adult offspring. However, a lot of molecular (RNA sequencing and gene expression) and histological studies were performed with newborn pups. Since a lot of processes happen differently in neonatal

and adult stages. It raises a concern whether this might affect the results and data interpretation.

We apologize for a lack of clarity on our part. Glucose homeostasis, insulin secretion, liver gluconeogenesis and testes biology were all studied in adult mice. This was done to determine if abnormal osteocalcin signaling during pregnancy could have long term consequences. It is only when we observed a phenotype in adult *Ocn*^{+/-} mice, that, to ascertain that it was of maternal and /or embryonic origin, we studied gene expression and β -cell proliferation in newborn mutant mice. These experiments were done to document that these phenotypes are, indeed, secondary to events occurring during pregnancy.

5. The authors conclude that osteocalcin regulates different genes and genetic pathways in organs where it signals through Gprc6a and in those like the brain where it signals through Gpr158 or Gpr3. The RNA seq results are not sufficient to prove that.

We agree with the reviewer and have modified the text accordingly.

Minor comments:

6. In figure 2E, the green and yellow bars need to be presented separately.

This has now been corrected.

Reviewer #3

There is a major issue with the manuscript that the authors must address. The authors state as fact that osteocalcin is a bone-derived hormone. However, 2 papers published in PLoS Genetics in 2020 using different osteocalcin knockout strains found no evidence of hormonal activity. The authors must discuss both these papers. Diegel et al, citation 33 in the current manuscript, did not observe differences in weight, adiposity, glucose levels, insulin levels, or testosterone levels in knockout offspring compared to wild-type offspring born to haplo-insufficient dams.

The reviewer raises an important point that needs to be analyzed thoroughly. Diegel et al. (Diegel et al, 2020), did state that osteocalcin deficiency did not affect blood glucose levels. However, if one looks at Figure 4B of their publication, the WT mice they present, have fed glucose levels that are well above 200mg/dl, and fasting blood glucose levels above 150mg/dl. These levels are, in both cases, higher than expected in WT mice of any genetic background exposed to the same fed/fasting conditions. Likewise, Diegel et al., (Diegel et al., 2020) may not have explicitly reported obesity, but the weight of their control mice is around **40g** with some of them peaking at **50g** (Diegel et al. (Diegel et al., 2020), Figure 4 A) and as we observed, higher than in the mutant mice. This body weight is higher than what it should be in 6-month-old mice that are predominantly on a 129Sv/ev genetic background. To the best of our knowledge, Diegel et al (Diegel et al., 2020), did not report circulating insulin levels, insulin secretion, liver gluconeogenesis, or gene expression. As for us, we measured ALL parameters of glucose metabolism, including circulating insulin levels, insulin secretion, liver gluconeogenesis and gene expression at two time points, newborn and adult. We also studied cognition and testicular steroidogenesis and gene expression as can be seen in Figures 1 (E), 2 (C-D), 3 (B, D, H, I, K) and 4 (B, E), and supplemental figures 1 (D, G, H, I) and 3 (B, G, H).

The results we obtained on larger sample size differ, but not substantially from those actually presented in the figures of Diegel et al. For instance, the obesity phenotype we observed in the WT mice in this strain shipped to us by Dr. Bart Williams (Figure 1) is *already noticeable* in Figure 4A of Diegel et al as mentioned above. Nevertheless, the discrepancies we observed when analyzing the same mutant mouse strain, were for us, like they are for the reviewer, a matter of concern.

This is why we approached another investigator previously unknown to us, Dr. Li, in another country (China) to study by himself and measure in his lab, cognition and static metabolic and endocrine parameters in another *Ocn*^{-/-} mice he generated through CRISPR/Cas9 technology (Qian et al., 2021). Our thinking was that regardless of the results Dr. Li would observe, it was worth to provide to the scientific community at large an analysis of three different *Ocn*-deficient mouse strains. One of them, performed by an investigator not involved in previous studies. This is the purpose of the experiments performed in the *Ocn*_{XL} mouse strain. Results in this strain have been presented in Figures 1 (F) and 2 (E-F). Dr. Li's group observed that as seen in *Ocn*_{GK} and *Ocn*_{BW} mouse strains, metabolic and cognitive functions are compromised in their mutant mice as well.

Therefore, before I could seriously consider the present manuscript for publication, the investigators must have glucose, insulin, and testosterone levels in wild-type and haplo-insufficient offspring of haplo-insufficient dams measured independently by a CRO (contract research organization). If these measures are validated by a CRO

We thank the reviewer for this suggestion. To address it, while remaining fully compliant with the NIH policy of use of federal funds, we contacted our program officer at the NIA, who considered the expense unjustified and in disagreement with the prudent use of federal funds.

This opinion being acknowledged, we note that most results obtained while studying mutant mice generated by Diegel et al. were reproduced in the laboratory of Dr. Li, in China, using for that purpose their own *Ocn-null* mouse strain. To add credence to results obtained in New York City and in Shenzhen, China, we now add in Figure 1 a summary of the behavioral analyses performed in New York city in a core facility and in Shenzhen by Dr. Li's group on *Ocn^{BW}-* and *Ocn^{XL}-* mouse strains respectively. In both cases there was a cognition deficit (Figure 1E, F).

Although they were by no means conducted for a fee and in a CRO, the independent assessment of cognition in a Core facility and most importantly, the location of Dr. Li's laboratory in another country, the facts that he uses a different *Ocn*^{-/-} mouse strain, and his relative inexperience in assessing glucose metabolism, are all assurances we believe, of his intellectual integrity.

Aside from the failure to cite and discuss work by others that do not support the main rationale or conclusions of the present paper, the investigators do not provide links to the whole genome and RNA sequencing datasets described in the manuscript's results. These datasets need to be available for reviewers to agree with the authors' conclusions.

We apologize to the reviewer. We are now submitting these results with the manuscript.

*I do not understand how the data in figures are presented. In the bar graphs, does each dot represent a single measure from every animal? If so, I would have imagined measures such as glucose, insulin, and testosterone would follow a normal distribution, particularly in strains that are isogenic (*Ocn^{GK}* and *Ocn^{XL}*). However, this doesn't appear to be the case for much of the presented data. For these reasons it would be helpful to include "(n=)" for every experiment and describe what individual dots represent in the figure legends.*

We now indicate *n* for each panel of each experiment. Each dot represents a single animal, this is now stated in figure legends.

*The investigators also need to explain why for an isogenic strain such as *Ocn^{GK}*, the mean fed glucose measurement in wild-type offspring from wild-type dams and haplo-insufficient sires is the same as that of haplo-insufficient offspring from knockout dams and wild-type sires (Figure 1C).*

The data the reviewer points out to were obtained initially from mice of very different ages (5 versus 12 weeks old). We have now performed anew this experiment using mice of the same age (12-weeks of age) and do not observe any significant differences.

Obesity is not a phenotype in Diegel et al (2020); instead weights of 6-month-old wild-type and knockout male and female littermates reported in Diegel et al (2020) are consistent with reference weights for C57BL6/J mice.

The obesity of the wild-type mice from the OCN_{BW} strain the reviewer refers to, was a surprise to us. After having observed it in every generation we went back to the paper of Diegel et al., (Diegel et al., 2020) and noted that it was present in their publication as well (Diegel et al., (Diegel et al., 2020) Figure 4A), therefore, we cannot consider it as new. We recognize this is an important issue but not necessarily one that should be addressed by us since in our hands $Ocn_{BW}^{-/-}$ have all the phenotypes observed in two other strains of $Ocn^{-/-}$ mice. Moreover, this would infringe on work Drs. Diegel and Williams may want to perform in mice they generated.

REFERENCES CITED

Benavides F, Rulicke T, Prins JB, Bussell J, Scavizzi F, Cinelli P, Herault Y, Wedekind D (2020) Genetic quality assurance and genetic monitoring of laboratory mice and rats: FELASA Working Group Report. *Lab Anim* 54: 135-148

Desbois C, Hogue DA, Karsenty G (1994) The mouse osteocalcin gene cluster contains three genes with two separate spatial and temporal patterns of expression. *J Biol Chem* 269

Diegel CR, Hann S, Ayturk UM, Hu JCW, Lim KE, Droscha CJ, Madaj ZB, Foxa GE, Izaguirre I, Transgenics Core VVA et al (2020) An osteocalcin-deficient mouse strain without endocrine abnormalities. *PLoS Genet* 16: e1008361

Oury F, Khimian L, Denny CA, Gardin A, Chamouni A, Goeden N, Huang YY, Lee H, Srinivas P, Gao XB et al (2013) Maternal and offspring pools of osteocalcin influence brain development and functions. *Cell* 155: 228-241

Pi M, Xu F, Ye R, Nishimoto SK, Williams RW, Lu L, Darryl Quarles L (2020) Role of GPRC6A in Regulating Hepatic Energy Metabolism in Mice. *Sci Rep* 10: 7216

Qian Z, Li H, Yang H, Yang Q, Lu Z, Wang L, Chen Y, Li X (2021) Osteocalcin attenuates oligodendrocyte differentiation and myelination via GPR37 signaling in the mouse brain. *Sci Adv* 7: eabi5811

Taberner C, Zolotukhin AS, Bear J, Schneider R, Karsenty G, Felber BK (1997) Identification of an RNA sequence within an intracisternal-A particle element able to replace Rev-mediated posttranscriptional regulation of human immunodeficiency virus type 1. *J Virol* 71: 95-101

Dear Prof. Karsenty

Thank you for the submission of your research manuscript to our journal. I apologize for the unusual delay in handling your manuscript. Your manuscript has been reviewed by another journal and you have submitted your manuscript together with the referee reports and your response to these to EMBO Reports. We have asked two independent experts in the field to act as arbitrator/advisor and to evaluate your manuscript and your response to the previous referees. Both advisors have meanwhile replied and both consider your data convincing and your response to the concerns of all three referees adequate. Given these positive evaluations, we have decided to proceed with the publication of your manuscript. I copy below the instructions how to format your manuscript for publication in EMBO Reports. Please also address the comments from advisor #2.

I am also happy to discuss the process further via e-mail or a video call, if you wish.

2) individual production quality figure files as .eps, .tif, .jpg (one file per figure). Please download our Figure Preparation Guidelines (figure preparation pdf) from our Author Guidelines pages <https://www.embopress.org/page/journal/14693178/authorguide> for more info on how to prepare your figures.

3) a complete author checklist, which you can download from our author guidelines (<<https://www.embopress.org/page/journal/14693178/authorguide>>). Please insert information in the checklist that is also reflected in the manuscript. The completed author checklist will also be part of the RPF.

4) Please note that all corresponding authors are required to supply an ORCID ID for their name upon submission of a revised manuscript (<<https://orcid.org/>>). Please find instructions on how to link your ORCID ID to your account in our manuscript tracking system in our Author guidelines (<<https://www.embopress.org/page/journal/14693178/authorguide#authorshipguidelines>>)

5) We replaced Supplementary Information with Expanded View (EV) Figures and Tables that are collapsible/expandable online. A maximum of 5 EV Figures can be typeset. EV Figures should be cited as 'Figure EV1, Figure EV2' etc... in the text and their respective legends should be included in the main text after the legends of regular figures.

6) Primary datasets (and computer code, where appropriate) produced in this study need to be deposited in an appropriate public database (see < <https://www.embopress.org/page/journal/14693178/authorguide#dataavailability>>). Specifically, we would kindly ask you to provide public access to the RNAseq dataset.

The accession numbers and database should be listed in a formal "Data Availability " section (placed after Materials & Method) that follows the model below (see also < <https://www.embopress.org/page/journal/14693178/authorguide#dataavailability>>). Please note that the Data Availability Section is restricted to new primary data that are part of this study.

Data availability

7) At EMBO Press we ask authors to provide source data for the main figures. Our source data coordinator will contact you to discuss which figure panels we would need source data for and will also provide you with helpful tips on how to upload and organize the files.

Additional information on source data and instruction on how to label the files are available
<<https://www.embopress.org/page/journal/14693178/authorguide#sourcedata>>.

8) The journal requires a statement specifying whether or not authors have competing interests (defined as all potential or actual interests that could be perceived to influence the presentation or interpretation of an article). In case of competing interests, this must be specified in your disclosure statement. Further information: <https://www.embopress.org/competing-interests>

9) Figure legends and data quantification:

- the name of the statistical test used to generate error bars and P values,
 - the number (n) of independent experiments (please specify technical or biological replicates) underlying each data point,
 - the nature of the bars and error bars (s.d., s.e.m.)
-
- If the data are obtained from n {less than or equal to} 5, show the individual data points in addition to the SD or SEM.
 - If the data are obtained from n {less than or equal to} 2, use scatter blots showing the individual data points.

10) Our journal encourages inclusion of *data citations in the reference list* to directly cite datasets that were re-used and obtained from public databases. Data citations in the article text are distinct from normal bibliographical citations and should directly link to the database records from which the data can be accessed. In the main text, data citations are formatted as follows: "Data ref: Smith et al, 2001" or "Data ref: NCBI Sequence Read Archive PRJNA342805, 2017". In the Reference list, data citations must be labeled with "[DATASET]". A data reference must provide the database name, accession number/identifiers and a resolvable link to the landing page from which the data can be accessed at the end of the reference. Further instructions are available at <<https://www.embopress.org/page/journal/14693178/authorguide#referencesformat>>.

11) As part of the EMBO publication's Transparent Editorial Process, EMBO Reports publishes online a Review Process File to accompany accepted manuscripts. This File will be published in conjunction with your paper and will include the comments from the advisor. The referee reports from the previous reviewers will not be published.

Kind regards,

Advisor #1:

Despite it is known that maternal health can affect the organismal homeostasis, little is known about the maternal factors leading to metabolic disturbances in the adults. Understanding of these factors are crucial to develop further therapies. In this sense, the current manuscript from Junior et al. is timely and identifies a highly interesting mechanism regarding the maternal osteocalcin's role in organismal homeostasis of the adult offspring. In this MS, authors, for the first time report that maternal osteocalcin synergize to regulate multiple physiological processes and contribute to establishing organismal homeostasis in adult offspring. They also show that osteocalcin uses Gprc6a in the periphery and signals through Grp158 and Gpr37 in the brain, which is a very novel and interesting observation.

I think the results presented in the manuscript are robust and novel and should be published as it is without a delay.

The manuscript has undergone peer review at another journal and I was also asked to comment on the reviewers' comments and the author's response to these. It looks clear to me that authors have responded scholarly and addressed the reviewers' points very well. It is kind of confusing that it was not accepted for publication in the other journal.

Reviewer 1 described the study "as very interesting manuscript reporting the complex functions and interactions of osteocalcin of maternal and embryonic origins on postnatal homeostasis in mice. The strengths of this study are the use of multiple strains of osteocalcin deficient mice, the characterization of several homeostatic functions regulated by osteocalcin and the molecular characterization of the impact of osteocalcin on these various physiological functions." and asked few points which need to be addressed.

Authors addressed all of the points of the reviewer 1.

Reviewer 2 indicated that this MS shows "maternal and embryonic osteocalcins synergize to regulate multiple physiological processes and thereby contribute to establishing organismal homeostasis in adult offspring. Overall, this study has recruited three different knockout mice models for osteocalcin and performed a big amount of experiments to test this hypothesis."

Authors also responded scholarly and addressed the points of this reviewer.

Reviewer 3, in my opinion, was out of line and his/her review was not scholarly performed. Reviewer C asked basic parameters such as glucose, insulin, and testosterone to be performed by a CRO independently. Dr. Karsenty's lab is in Columbia University in a very serious department.. Measuring glucose or insulin or testosterone levels are the simplest experimental procedures and a reviewer should not ask those parameters to be checked by a CRO. On the other hand the authors has responded scholarly to this reviewer's comments and addressed this reviewer's points too.

Advisor #2:

The manuscript by Yadev et al, addresses an important and controversial topic of bone biology related to the hormonal actions of the osteoblast-derived protein osteocalcin (OCN). Prior work by the Karsenty lab, but importantly also by other groups (1,2), has provided strong evidence that OCN produced by osteoblasts acts on multiple end organs to regulate diverse functions ranging from bone mass accumulation to body weight, adiposity, glucose and energy metabolism, male fertility, and cognition. Subsequent findings from the Karsenty lab suggest that some OCN actions manifest during pregnancy via maternal fetal transfer. For years, this proposed "hormonal" role for has been challenged by some in the field often without tangible evidence to the contrary. However, two papers published together in Plos Genetics (3,4) conducted in independent labs using different osteocalcin knockout strains found no differences in weight, adiposity, glucose levels, insulin levels, or testosterone levels in knockout offspring compared to wild-type offspring born to haplo-insufficient dams. The results from these studies directly challenge the hormonal concept of OCN function.

It is in this background that the study by Yadev et al was conducted, which further explores the maternal-fetal axis in OCN biology. Specifically, they seek to test the hypothesis that maternal and embryonic OCN synergize to regulate multiple physiological processes and thereby contribute to establishing organismal homeostasis in adult offspring. The results suggest that haploinsufficiency in a pregnant mother affects glucose levels, insulin response, pancreatic beta cell development, and testis development in haploinsufficient and/or knockout offspring but not in wild-type offspring.

The paper has undergone a rigorous peer review by three reviewers at another highly respected journal. The authors have addressed many of the reviewers' points in some cases performing new experiments to address the concerns. In my opinion, these responses have satisfactorily addressed the concerns of Reviewers 1 and 2. Reviewer 2 also questioned the overall novelty of the study given the 2014 paper by the group on the maternal effects of osteocalcin (5) and was concerned about the difficulty in comparison of the current data to previous work given the different genetic backgrounds studied. In this regard, I think the authors could do a better job in describing the motivation for the current study (see below).

The most serious points of contention come from Reviewer 3 who focuses on the two recent Plos Genetics papers and their failure to find metabolic and reproductive phenotypes that would support the hormonal role of OCN. Yadev argue quite convincingly that certain aspects of one of the papers (Diegel et al) are suspect e.g. non-fasting high glucose levels, possible obesity, etc. I also noted these discrepancies when it appeared and in general this paper lacks rigor which is especially problematic since its main purpose appears to have been to discredit the concept for a metabolic role of ONN. Reviewer 3 goes

on to call for independent verification of key biochemical data in the Yadev manuscript stating: "Therefore, before I could seriously consider the present manuscript for publication, the investigators must have glucose, insulin, and testosterone levels in wild-type and haploinsufficient offspring of haploinsufficient dams measured independently by a CRO (contract research organization). I find this type of ad hoc demand highly unorthodox, without precedent and inappropriate in the context of a manuscript review.

In conclusion, upon careful review of the paper's original findings and the critiques from the prior review, I recommend the paper for publication without additional studies but suggest that the authors consider the following editorial points:

1. Add a section to the Introduction that better rationalizes the importance of the study with respect to both the hormonal and maternal fetal aspects of ONN.
2. Bolster the paragraph in the Discussion to discuss the results of the two PLOS Genetics papers more thoroughly in the context of their present findings.

References:

1. Fulzele K, Riddle RC, DiGirolamo DJ, Cao X, et al. Insulin receptor signaling in osteoblasts regulates postnatal bone acquisition and body composition. *Cell*. 2010;142:309-19.
2. Tara C. Brennan-Speranza, Holger Henneicke, Sylvia J. Gasparini, Katharina I. Blankenstein, Uta Heinevetter, Victoria C. Cogger, Dmitri Svistounov, Yaqing Zhang, Gregory J. Cooney, Frank Buttgereit, Colin R. Dunstan, Caren Gundberg, Hong Zhou, Markus J. Seibel Osteoblasts mediate the adverse effects of glucocorticoids on fuel metabolism *J Clin Invest*. 2012;122(11):4172-4189
3. Diegel CR, Hann S, Ayturk UM, Hu JCW, Lim KE, Droscha CJ, Madaj ZB, Foxa GE, Izaguirre I, Transgenics Core VVA et al (2020) An osteocalcin-deficient mouse strain without endocrine abnormalities. *PLoS Genet* 16: e1008361
4. Moriishi T, Ozasa R, Ishimoto T, Nakano T, Hasegawa T, et al. (2020) Osteocalcin is necessary for the alignment of apatite crystallites, but not glucose metabolism, testosterone synthesis, or muscle mass. *PLoS Genetics*. 16:e1008586.
5. Oury F, Khrimian L, Denny CA, Gardin A, Chamouni A, Goeden N, Huang YY, Lee H, Srinivas P, Gao XB, Suyama S, Langer T, Mann JJ, Horvath TL, Bonnin A, Karsenty G. Maternal and offspring pools of osteocalcin influence brain development and functions. *Cell*. 2013 Sep 26;155(1):228-41

The authors have addressed all minor editorial requests.

Dear Gerard,

Thank you for the submission of your revised manuscript to our journal. We have meanwhile completed all editorial checks and noted that the following points still need your attention before we can proceed with the official acceptance of your manuscript.

- Please provide all figures in portrait orientation (and at higher resolution, if possible).
- Please provide up to 5 keywords
- We note that your study has 4 co-first and 5 co-corresponding authors. Please provide a justification for these shared authorships.
- Please correct the heading "Data and materials availability" to "Data availability". Please remove the statement "All data are available in the main text or the supplementary materials; mouse strains are all publicly available" and only refer to the RNAseq dataset. Please provide the accession ID and a link that resolves directly to the deposited dataset. If you consider the statement "mouse strains are all publicly available" important, it can also remain.
- Please remove the Author Contributions from the manuscript file and make sure that the author contributions in our online submission system are correct and up-to-date. The information you specified in the system will be automatically retrieved and typeset into the article. You can enter additional information in the free text box provided, if you wish.
- Please update the 'Conflict of interest' paragraph to our new 'Disclosure and competing interests statement'. For more information see <https://www.embopress.org/page/journal/14693178/authorguide#conflictsofinterest>
- On page 5 you refer to "unpublished observation". Please note that our editorial policies require that all statements and conclusions are supported by the corresponding data. Therefore, either remove the observation or provide the corresponding, supporting data.
- Page 16 seems to contain a typo. Should "The causes of these differences are presently known...." read "...are presently unknown"?
- Our production/data editors have asked you to clarify several points in the figure legends (see below). Please incorporate these changes in the manuscript and return the revised file with tracked changes with your final manuscript submission.
 1. Please note that a separate 'Data Information' section is required in the legends of Figure 1-4, 5. This means that e.g. the statement "In bar plots, each dot represents an individual mouse. All data are shown as mean {plus minus} SEM. Statistical significance was determined by one-way Kruskal-Wallis test followed by post hoc multiple comparisons test. * $p < 0.05$, ** $p < 0.01$, *** $p < 0.001$, **** $p < 0.0001$." in Figure 2 is preceded by "Data information:" as this description applies to all panels in the figure.
 2. Please note that the Expanded View Figures miss EV2 and EV5. Please correct the labels to Figure EV1-4 and the callouts in the text.
 3. Please note that in figures 1e-f; 2a-f there is a mismatch between the annotated p values in the figure legend and the annotated p values in the figure file that should be corrected.
 4. Please indicate the statistical test used for data analysis in the legends of figures 5b; 7b.
 5. Please note that information related to n is missing in the legends of figures 5b; 7b.
 6. Please note that the scale bar is missing for figure EV1a
- The heading "Methods" needs correcting to Materials and Methods
- I had already informed you that we require source data for all figures and I list below again, which data we require:
 - a) Figure 1E, F: quantification as .xls or .csv file
 - b) Figure 1C: micr. image
 - c) Figure 2A-F: quantification as .xls or .csv file
 - d) Figure 3A-K: quantification as .xls or .csv file
 - e) Figure 4A, B, D, E: : quantification as .xls or .csv file
 - f) Figure 4C: micr. Image
 - g) Figure 6C, D, E: micr. Image
 - h) Figure 6A, B, F, G: quantification as .xls or .csv file

Similar source data for EV figures is recommended but not mandatory.

Please find instructions for the organization of source data in the attached document.

- Finally, EMBO Reports papers are accompanied online by A) a short (1-2 sentences) summary of the findings and their

significance, B) 2-3 bullet points highlighting key results and C) a synopsis image that is 550x300-600 pixels large (width x height) in PNG or JPG format. You can either show a model or key data in the synopsis image. Please note that the size is rather small and that text needs to be readable at the final size. Please send us this information along with the revised manuscript.

I am looking forward to receive your manuscript once it is ready.

Kind regards,

Martina

All editorial and formatting issues were resolved by the authors.

Prof. Gerard Karsenty
Columbia university medical center
Dept. of Genetics and development
701W 168th street New York, NY, 10032
Hammer building, Room 1602A
New york city, NY 10032
United States

Dear Gerard,

We have meanwhile checked the revised manuscript files and I am very pleased to accept your manuscript for publication in the next available issue of EMBO reports. Thank you for your contribution to our journal.

Kind regards,

Martina
